# Synthesis of High-Input Impedance Electronically Tunable Voltage-Mode Second-Order Low-Pass, Band-Pass, and High-Pass Filters Based on LT1228 Integrated Circuits

**DOI:** 10.3390/s22239379

**Published:** 2022-12-01

**Authors:** Hua-Pin Chen, Shih-Jun Chen, Chih-Yang Chang

**Affiliations:** Department of Electronic Engineering, Ming Chi University of Technology, Taipei 24301, Taiwan

**Keywords:** filters, analog circuit design, integrated circuit, commercial LT1228

## Abstract

This paper introduces two new high-input impedance electronically tunable voltage-mode (VM) multifunction second-order architectures with band-pass (BP), low-pass (LP), and high-pass (HP) filters. Both proposed architectures have one input and five outputs, implemented employing three commercial LT1228 integrated circuits (ICs), two grounded capacitors, and five resistors. Both proposed architectures also feature one high-impedance input port and three low-impedance output ports for easy connection to other VM configurations without the need for VM buffers. The two proposed VM LT1228-based second-order multifunction filters simultaneously provide BP, LP, and HP filter transfer functions at V_o1_, V_o2_, and V_o3_ output terminals. The pole angular frequencies and the quality factors of the two proposed VM LT1228-based second-order multifunction filters can be electronically and orthogonally adjusted by the bias currents from their corresponding commercial LT1228 ICs, and can be independently adjusted in special cases. In addition, both proposed VM LT1228-based second-order multifunction filters have two independent gain-controlled BP and LP filter transfer functions at V_o4_ and V_o5_ output terminals, respectively. Based on the three commercial LT1228 ICs and several passive components, simulations and experimental measurements are provided to verify the theoretical predictions and demonstrate the performance of the two proposed high-input impedance electronically tunable VM LT1228-based second-order multifunction filters. The measured input 1-dB power gain compression point (P1dB), third-order IMD (IMD3), third-order intercept (TOI) point, and spurious-free dynamic range (SFDR) of the first proposed filter were −7.1 dBm, −48.84 dBc, 4.133 dBm, and 45.02 dBc, respectively. The measured input P1dB, IMD3, TOI, and SFDR of the second proposed filter were −7 dBm, −49.65 dBc, 4.316 dBm, and 45.88 dBc, respectively. Both proposed filters use a topology synthesis method based on the VM second-order non-inverting/inverting HP filter transfer functions to generate the BP, LP and HP filter transfer functions simultaneously, making them suitable for applications in three-way crossover networks.

## 1. Introduction

Electronically tunable active filters and oscillators designed with active components are widely used in sensor applications such as electrocardiography systems [1], biosensors [2], electronically tunable LC oscillators [3], and phase-sensitive detection [4]. Especially in the electronic sensor systems, electronically tunable active filters are used to filter out noise in the sensor systems [5]. An interesting dual-output MOSFET-only filter without any passive components is also presented in [6]. It has proven to be an effective solution for high frequency ranges. Electronically tunable voltage-mode (VM) high-pass filter (HPF), band-pass filter (BPF), and low-pass filter (LPF) topologies suitable for integrated circuit (IC) structures have been a constant endeavor of circuit designers, and have become very important architecture circuits in sensors, analog systems, electrical and electronic engineering works. The choice of active building blocks (ABBs) plays an important role in analog filter design because one expects VM second-order multifunction filters with high input and low output impedances to allow cascading of the VM circuits. Therefore, many circuit architecture studies use different high-performance ABBs to implement different kinds of filters and oscillators [7,8,9,10,11,12,13,14,15,16,17,18,19,20,21,22]. In [22], a high-input impedance VM second-order filter based on four second-generation current conveyor (CCII) active components and six passive components is proposed. The filter can simultaneously implement LP, BP, HP, band-reject (BR), and all-pass (AP) filters from the same circuit configuration, but the filter lacks the low-output impedances. In [23,24], three current feedback amplifier (CFA) active components and six passive components are used in each VM second-order filter circuit design. The technique of [23,24] is implemented with three commercially available AD844 ICs, but neither the pole angular frequency (ω_o_) nor the quality factor (Q) can be electronically adjusted. In [25], four CFA active components, seven/eight passive components, and one/two switches (SWs) are used in each VM second-order filter circuit design. Each circuit in [25] is implemented using four commercially available AD844 ICs, allowing four different filter responses simultaneously, but the parameters ω_o_ and Q of each circuit still cannot be electronically adjusted. In [26,27], three voltage differencing differential difference amplifier (VDDDA) active components and three passive components are used in the VM second-order filter circuit design. Each technology has the ability to electronically adjust ω_o_ and Q, enabling simultaneous implementation of LP, BP, HP, BR, and AP filters, but each circuit requires six commercial ICs to implement the VM filter architecture, namely three LM13700 and three AD830 ICs. In [28], two voltage differencing differential input buffered amplifier (VD-DIBA) active components and four passive components are used in the VM second-order filter circuit design. The circuit also has the ability to electronically adjust ω_o_ and Q, enabling simultaneous implementation of LP, BP, HP, and BR filters, but the circuit is implemented using four commercial ICs, namely two LM13700 and two AD830 ICs. Two electronically tunable VM second-order filters based on five operational transconductance amplifiers (OTAs) have been designed and developed [29,30], but these two configurations suffer from low-output impedance and the use of five LT1228 ICs. In [31], four OTAs and two grounded capacitors are used in the design of an electronically tunable VM second-order filter, but the technique is still implemented using five commercially available LT1228 ICs.

Due to its wide transconductance amplifier, high voltage-gain, large signal bandwidth, wide supply voltage, high accuracy, and high drive capability, the commercially available LT1228 IC is an interesting electronically tunable active component suitable for many circuit designs. As a result, many active VM second-order filters, oscillators, inductance simulators, and wave generators based on the attractive commercial LT1228 ICs were proposed in the literature [32,33,34,35,36,37,38,39,40,41,42,43,44,45,46,47,48]. The LT1228 implements gain-controlled through a transconductance g_m_ at the front end and a CFA at the back end, and combines these two amplifiers into an 8-pin package [49]. It operates on any supply voltage between 4 V (±2 V) and 30 V (±15 V) [49]. The LT1228’s first-stage g_m_ transconductance amplifier has a high-impedance differential input pair and a high-impedance current output to provide a wide range of voltage-to-current conversion. The LT1228’s second stage CFA has a low-output impedance and a wide voltage-gain range, making it ideal for driving low-impedance loads and avoiding loading effects. According to [32], two interesting independent amplitude VM BPFs are proposed, but both the HPF and LPF cannot be simultaneously obtained in each circuit configuration. The high-fidelity three-way speaker system uses one HPF, one BPF, and one LPF to connect the tweeter, midrange, and woofer, so the HPF, BPF, and LPF structures must be simultaneously implemented in the circuit design [31,33]. An interesting LT1228-based electronically tunable VM second-order multifunction filter is presented in [33], which simultaneously implements HPF, BPF, and LPF transfer functions from the same circuit configuration. According to [33], the circuit has the following features: (i) The filter can simultaneously generate VM second-order HPF, BPF, and LPF transfer functions, and is suitable for three-way crossover networks. (ii) The parameters of the filter, ω_o_ and Q, permit for electronic and orthogonal controllability. (iii) The HPF and BPF responses provide low-output impedance, and can be directly cascaded to other VM circuits without the use of additional VM buffers.

In this paper, two new synthesis methods for VM LT1228-based second-order multifunction filters based on non-inverting/inverting HPF transfer functions are designed and developed. Each of the two proposed designed and synthesized VM second-order multifunction filters uses three commercial LT1228s, five resistors, and two grounded capacitors, which is one more resistor than the filter topology implemented in [33], to achieve two independent gain-controlled filter responses. Both the designed and the synthesized VM second-order multifunction filters have the following advantages.

(i)The filter can simultaneously generate HPF, BPF, and LPF second-order transfer functions, and is suitable for three-way crossover networks.(ii)The parameters of the filter, ω_o_ and Q, permit for electronic and orthogonal controllability.(iii)The filter parameter Q has independent and electronic tuning capability.(iv)The filter provides a high-impedance input suitable for cascading voltage input stages.(v)The HPF, BPF, and LPF responses provide low-impedance outputs suitable for cascading voltage output stages.(vi)Passive components do not require matching conditions.(vii)The passband gains of the LPF and BPF responses can be controlled effectively and independently without affecting the filter parameters ω_o_ and Q.(viii)Synthesis methods of the filter topologies based on VM non-inverting/inverting HPF second-order transfer functions.

In addition to the three advantages (i) to (iii) realized in [33], these two new circuits provide the functions (iv) to (viii). In Table 1, the main characteristics of the two proposed LT1228-based electronically tunable VM second-order multifunction filters are compared with previous VM second-order multifunction filters implemented using commercially available ICs technology. As shown in Table 1, the two proposed LT1228-based electronically tunable VM second-order multifunction filters have independent gain-controlled LPF and BPF functions and satisfy all the main (i) to (viii) advantages. Both proposed filters use a topology synthesis method based on the VM second-order non-inverting/inverting HPF transfer functions to generate the BPF, LPF, and HPF transfer functions simultaneously, making them suitable for applications in three-way crossover networks [33]. To confirm the circuit performances of both proposed VM multifunction filters, PSpice simulation, measurement and theoretical calculation results are performed using three LT1228 ICs and several passive components.

## 2. Theory and Implementation of VM Second-Order Multifunction Filters Based on Non-Inverting/Inverting HPF Transfer Functions

### 2.1. First Proposed Synthesis Principle and Analysis Theory Based on Inverting HPF Transfer Function

To synthesize the first proposed VM second-order multifunction filter design system block, the VM inverting HPF (IHPF) second-order transfer function with two integrator time constants τ_1_ and τ_2_ and a voltage passband gain k_1_ can be considered as the following function.
(1)$$\frac{V_{\mathrm{IHPF}}}{V_{\mathrm{in}}}=\frac{-k_{1}s^{2}}{s^{2}+s\frac{k_{1}}{\tau_{1}}+\frac{1}{\tau_{1}\tau_{2}}}$$
where V_IHPF_ is one of the output voltages of the first proposed filter, and V_in_ is the input voltage signal of the circuit.

To decompose the VM IHPF second-order transfer function into the VM non-inverting LPF (NLPF) and non-inverting BPF (NBPF) transfer functions, the terms of Equation (1) can be cross-multiplied first, and then divided by s^2^ to obtain
(2)$$V_{\mathrm{IHPF}}=-\frac{k_{1}}{{s\tau }_{1}}V_{\mathrm{IHPF}}-\frac{1}{s^{2}\tau_{1}\tau_{2}}V_{\mathrm{IHPF}}-k_{1}V_{\mathrm{in}}$$

Rearranging Equation (2) can be rewritten as
(3)$$V_{\mathrm{IHPF}}={(k}_{1}+\frac{1}{{s\tau }_{2}})(-\frac{1}{{s\tau }_{1}}V_{\mathrm{IHPF}})-k_{1}V_{\mathrm{in}}$$

Let
(4)$$V_{\mathrm{NBPF}}=-\frac{1}{{s\tau }_{1}}V_{\mathrm{IHPF}}$$

This indicates that the NBPF signal can be achieved by cascading the IHPF signal with an inverting loss integrator, and Equation (3) becomes
(5)$$V_{\mathrm{IHPF}}=k_{1}V_{\mathrm{NBPF}}+\frac{1}{{s\tau }_{2}}V_{\mathrm{NBPF}}-k_{1}V_{\mathrm{in}}$$

Let
(6)$$V_{\mathrm{NLPF}}=\frac{1}{{s\tau }_{2}}V_{\mathrm{NBPF}}$$

This means that the NLPF signal can be achieved by cascading the NBPF signal with a non-inverting loss integrator, and Equation (6) becomes
(7)$$V_{\mathrm{IHPF}}=k_{1}V_{\mathrm{NBPF}}+V_{\mathrm{NLPF}}-k_{1}V_{\mathrm{in}}$$

According to Equations (4), (6), and (7), the first proposed VM second-order multifunction filter system structure can be synthesized. The output signals of NLPF and NBPF, and the first filter parameters of ω_o_ and Q can be derived as follows.
(8)$$\frac{V_{\mathrm{NBPF}}}{V_{\mathrm{in}}}=\frac{s\frac{k_{1}}{\tau_{1}}}{s^{2}+s\frac{k_{1}}{\tau_{1}}+\frac{1}{\tau_{1}\tau_{2}}}$$
(9)$$\frac{V_{\mathrm{NLPF}}}{V_{\mathrm{in}}}=\frac{k_{1}\frac{1}{\tau_{1}\tau_{2}}}{s^{2}+s\frac{k_{1}}{\tau_{1}}+\frac{1}{\tau_{1}\tau_{2}}}$$
(10)$$\omega_{o}=\sqrt{\frac{1}{\tau_{1}\tau_{2}}}$$
(11)$$Q=\frac{1}{k_{1}}\sqrt{\frac{\tau_{1}}{\tau_{2}}}$$

Equations (10) and (11) indicate that the parameter Q can be independently tuned by adjusting the voltage gain building block of k_1_ without affecting the parameter ω_o_. In the special case of τ_1_ = τ_2_ = τ, the parameter ω_o_ can also be independently tuned by τ without affecting the parameter Q. Based on the DC bias current I_B_ of the corresponding LT1228, the first proposed filter parameters of ω_o_ and Q can be controlled electronically and orthogonally.

In Equation (1), the VM IHPF second-order transfer function is decomposed into the VM NLPF and NBPF second-order transfer functions, and the synthesis of the system block diagram can be achieved. Equations (4), (6), and (7) can be rearranged as Equation (12) in the form of an input–output matrix, and the system block diagram synthesis for the first proposed VM LT1228-based filter design is shown in Figure 1. It uses one input voltage node and three output voltage nodes, and consists of a non-inverting lossless integrator, an inverting lossless integrator, and a voltage gain building block.
(12)$$\left[\begin{array}{ccc} 1 & 0 & \frac{1}{{s\tau }_{1}}\\ -\frac{1}{{s\tau }_{2}} & 1 & 0\\ -k_{1} & -1 & 1 \end{array}\right]\left[\begin{array}{c} V_{\mathrm{NBPF}}\\ V_{\mathrm{NLPF}}\\ V_{\mathrm{IHPF}} \end{array}\right]=\left[\begin{array}{c} 0\\ 0\\ -k_{1}V_{\mathrm{in}} \end{array}\right]$$

In Figure 1, taking the system blocks of τ_1_ = C_1_/g_m1_, τ_2_ = C_2_/g_m2_ and k_1_ = g_m3_R, Equation (12) can be rewritten as
(13)$$\left[\begin{array}{ccc} 1 & 0 & \frac{g_{m1}}{{\mathrm{sC}}_{1}}\\ -\frac{g_{m2}}{{\mathrm{sC}}_{2}} & 1 & 0\\ -g_{m3}R & -1 & 1 \end{array}\right]\left[\begin{array}{c} V_{\mathrm{NBPF}}\\ V_{\mathrm{NLPF}}\\ V_{\mathrm{IHPF}} \end{array}\right]=\left[\begin{array}{c} 0\\ 0\\ -g_{m3}{\mathrm{RV}}_{\mathrm{in}} \end{array}\right]$$
where C_1_ and C_2_ are two capacitors, g_m1_, g_m2_, and g_m3_ are three LT1228 transconductance amplifiers, and R is a resistor. Solving Equation (13), the parameters ω_o_ and Q of the first filter and the three VM second-order transfer functions of the NBPF, NLPF and IHPF can be obtained as follows.
(14)$$\frac{V_{\mathrm{NBPF}}}{V_{\mathrm{in}}}=\frac{{\mathrm{sC}}_{2}g_{m1}g_{m3}R}{s^{2}C_{1}C_{2}+{\mathrm{sC}}_{2}g_{m1}g_{m3}R+g_{m1}g_{m2}}$$
(15)$$\frac{V_{\mathrm{NLPF}}}{V_{\mathrm{in}}}=\frac{g_{m1}g_{m2}g_{m3}R}{s^{2}C_{1}C_{2}+{\mathrm{sC}}_{2}g_{m1}g_{m3}R+g_{m1}g_{m2}}$$
(16)$$\frac{V_{\mathrm{IHPF}}}{V_{\mathrm{in}}}=\frac{-s^{2}C_{1}C_{2}g_{m3}R}{s^{2}C_{1}C_{2}+{\mathrm{sC}}_{2}g_{m1}g_{m3}R+g_{m1}g_{m2}}$$
(17)$$\omega_{o}=\sqrt{\frac{g_{m1}g_{m2}}{C_{1}C_{2}}}$$
(18)$$Q=\frac{1}{g_{m3}R}\sqrt{\frac{C_{1}g_{m2}}{C_{2}g_{m1}}}$$

Equations (17) and (18) indicate that the parameter Q can be independently tuned by adjusting g_m3_ and/or R without affecting the parameter ω_o_. In the special case of C_1_ = C_2_, and g_m1_ = g_m2_= g_m_, the parameter ω_o_ can also be independently tuned by g_m_ without affecting the parameter Q. Based on the DC bias current I_B_ of the corresponding LT1228, the filter parameters of ω_o_ and Q can be controlled electronically and orthogonally. Based on Figure 1 and Equation (13), Section 2.2 discusses the first proposed high-input impedance electronically tunable VM one-input five-output second-order multifunction filter based on three LT1228 ICs.

### 2.2. First Proposed VM LT1228-Based Second-Order Multifunction Filter

The LT1228 contains a g_m_ transconductance amplifier with a DC bias current I_B_ and a wide range of voltage-gain [33,49]. Figure 2a,b show the circuit symbol for the commercial LT1228 IC and the package for the 8-pin IC configuration, respectively. In Figure 2b, the g_m_ transconductance differential voltages of V+ and V− have high-input impedance, the current output at Y terminal has high-output impedance, and the voltage at X terminal follows the voltage at Y terminal. The W terminal voltage has good linearity and can be directly connected to an external low resistance value. Therefore, the W terminal of the commercial LT1228 IC is suitable for driving low-impedance loads such as loudspeakers and cables. The supply voltages of V_CC_ and V_EE_ are the positive and negative supply voltages of the commercial LT1228 IC. The port factors of the LT1228 with high-input impedance V+ and V−, low-output impedance W and high-output impedance Y are important in the design of any analog circuits. Figure 3 shows the equivalent circuit of a commercial LT1228 IC, whose ideal property relations can be described by [32,33]:(19)$$\left[\begin{array}{c} I_{v+}\\ I_{v-}\\ I_{y}\\ V_{x}\\ V_{w} \end{array}\right]=\left[\begin{array}{ccccc} 0 & 0 & 0 & 0 & 0\\ 0 & 0 & 0 & 0 & 0\\ g_{m} & -g_{m} & 0 & 0 & 0\\ 0 & 0 & 1 & 0 & 0\\ 0 & 0 & 0 & R_{T} & 0 \end{array}\right]\left[\begin{array}{c} V_{+}\\ V_{-}\\ V_{y}\\ I_{x}\\ I_{w} \end{array}\right]$$
where R_T_ is the transresistance gain of LT1228 and ideal value of R_T_ is close to infinity. The transconductance g_m_ of LT1228 depends on the I_B_ and the g_m_-value can be described as [32,49]
(20)$$g_{m}={10I}_{B}=\frac{V_{\mathrm{CC}}-V_{\mathrm{EE}}-{2V}_{\mathrm{BE}}}{R_{B}}$$
where R_B_ is the bias current control resistor.

Based on the synthesis of the system block diagram in Figure 1, the first proposed VM LT1228-based second-order multifunction filter configuration using three commercial LT1228s, five resistors, and two capacitors connected to the ground is shown in Figure 4. The nodal analysis of the first proposed VM LT1228-based second-order multifunction filter configuration can be written as follows.
(21)$$V_{o1}=-\frac{g_{m1}}{{\mathrm{sC}}_{1}}V_{o3}$$
(22)$$V_{o2}=\frac{g_{m2}}{{\mathrm{sC}}_{2}}V_{o1}$$
(23)$$V_{o3}=g_{m3}{R(V}_{o1}-V_{\mathrm{in}})+V_{o2}$$
(24)$$V_{o4}=(1+\frac{R_{1}}{R_{2}}{)V}_{o1}$$
(25)$$V_{o5}=(1+\frac{R_{3}}{R_{4}}{)V}_{o2}$$

Equations (21)–(23) can be rearranged in the form of an input–output matrix equation as follows.
(26)$$\left[\begin{array}{ccc} 1 & 0 & \frac{g_{m1}}{{\mathrm{sC}}_{1}}\\ -\frac{g_{m2}}{{\mathrm{sC}}_{2}} & 1 & 0\\ -g_{m3}R & -1 & 1 \end{array}\right]\left[\begin{array}{c} V_{o1}\\ V_{o2}\\ V_{o3} \end{array}\right]=\left[\begin{array}{c} 0\\ 0\\ -g_{m3}{\mathrm{RV}}_{\mathrm{in}} \end{array}\right]$$

Equation (26) is the same as Equation (13), if we let the outputs V_o1_ = V_NBPF_, V_o2_ = V_NLPF_, and V_o3_ = V_IHPF_. Therefore, the first proposed VM LT1228-based second-order multifunction filter proposed in Figure 4 simultaneously provides the NBPF, NLPF, and IHPF transfer functions at the V_o1_, V_o2_, and V_o3_ outputs, respectively, as shown in Equations (27)–(29).
(27)$$\frac{V_{o1}}{V_{\mathrm{in}}}=\frac{{\mathrm{sC}}_{2}g_{m1}g_{m3}R}{s^{2}C_{1}C_{2}+{\mathrm{sC}}_{2}g_{m1}g_{m3}R+g_{m1}g_{m2}}$$
(28)$$\frac{V_{o2}}{V_{\mathrm{in}}}=\frac{g_{m1}g_{m2}g_{m3}R}{s^{2}C_{1}C_{2}+{\mathrm{sC}}_{2}g_{m1}g_{m3}R+g_{m1}g_{m2}}$$
(29)$$\frac{V_{o3}}{V_{\mathrm{in}}}=\frac{-s^{2}C_{1}C_{2}g_{m3}R}{s^{2}C_{1}C_{2}+{\mathrm{sC}}_{2}g_{m1}g_{m3}R+g_{m1}g_{m2}}$$

Based on Equations (27)–(29), the first proposed filter parameters of ω_o_ and Q are the same as Equations (17) and (18). According to Equations (24) and (25), the first proposed VM LT1228-based second-order multifunction filter has two independent gain-controlled BPF and LPF transfer functions at V_o4_ and V_o5_ output terminals, respectively, as shown in Equations (30) and (31).
(30)$$\frac{V_{o4}}{V_{\mathrm{in}}}=(1+\frac{R_{1}}{R_{2}})\frac{{\mathrm{sC}}_{2}g_{m1}g_{m3}R}{s^{2}C_{1}C_{2}+{\mathrm{sC}}_{2}g_{m1}g_{m3}R+g_{m1}g_{m2}}$$
(31)$$\frac{V_{o5}}{V_{\mathrm{in}}}=(1+\frac{R_{3}}{R_{4}})\frac{g_{m1}g_{m2}g_{m3}R}{s^{2}C_{1}C_{2}+{\mathrm{sC}}_{2}g_{m1}g_{m3}R+g_{m1}g_{m2}}$$

In Equations (30) and (31), the first filter has two independent gain-controlled BPF and LPF transfer functions at the V_o4_ and V_o5_ outputs, and its four resistors of R_1_, R_2_, R_3_ and R_4_ can be tuned to independent gain control without affecting the design parameters of ω_o_ and Q. In Figure 4, the three output voltage nodes V_o3_, V_o4_, and V_o5_ are connected to the corresponding W terminal of each LT1228 to provide low output impedance. If C_1_ = C_2_ = C, g_m1_ = g_m2_ = 10I_B1_ and g_m3_ = 10I_B3_, the parameters ω_o_ and Q in Equations (17) and (18) can be rewritten as
(32)$$\omega_{o}=\frac{{10I}_{B1}}{C}$$
(33)$$Q=\frac{1}{{10I}_{B3}R}$$

In this particular case, the parameter ω_o_ can be tuned electronically and independently by the bias current I_B1_ of LT1228 without affecting the parameter Q, and the parameter Q can also be tuned electronically and independently by the bias current I_B3_ of LT1228 without affecting the parameter ω_o_.

### 2.3. Second Proposed Synthesis Principle and Analysis Theory Based on Non-Inverting HPF Transfer Function

To synthesize the second proposed VM second-order multifunction filter design system block, the VM non-inverting HPF (NHPF) second-order transfer function with two integrator time constants τ_3_ and τ_4_ and a voltage passband gain k_2_ can be considered as the following function.
(34)$$\frac{V_{\mathrm{NHPF}}}{V_{\mathrm{in}}}=\frac{k_{2}s^{2}}{s^{2}+s\frac{k_{2}}{\tau_{3}}+\frac{1}{\tau_{3}\tau_{4}}}$$
where V_NHPF_ is one of the output voltages of the second proposed filter, and V_in_ is the input voltage signal of the circuit

To decompose the VM NHPF second-order transfer function into the VM inverting LPF (ILPF) and NBPF transfer functions, the terms of Equation (34) can be cross-multiplied first, and then divided by s^2^ to obtain
(35)$$V_{\mathrm{NHPF}}=-\frac{k_{2}}{{s\tau }_{3}}V_{\mathrm{NHPF}}-\frac{1}{s^{2}\tau_{3}\tau_{4}}V_{\mathrm{NHPF}}+k_{2}V_{\mathrm{in}}$$

Rearranging Equation (35) can be rewritten as
(36)$$V_{\mathrm{NHPF}}=-{(k}_{2}+\frac{1}{{s\tau }_{4}})(\frac{1}{{s\tau }_{3}}V_{\mathrm{NHPF}})+k_{2}V_{\mathrm{in}}$$

Let
(37)$$V_{\mathrm{NBPF}}=\frac{1}{{s\tau }_{3}}V_{\mathrm{NHPF}}$$

This indicates that the NBPF signal can be achieved by cascading the NHPF signal with a non-inverting loss integrator, and Equation (37) becomes
(38)$$V_{\mathrm{NHPF}}=-k_{2}V_{\mathrm{NBPF}}-\frac{1}{{s\tau }_{4}}V_{\mathrm{NBPF}}+k_{2}V_{\mathrm{in}}$$

Let
(39)$$V_{\mathrm{ILPF}}=-\frac{1}{{s\tau }_{4}}V_{\mathrm{NBPF}}$$

This means that the ILPF signal can be achieved by cascading the NBPF signal with an inverting loss integrator, and Equation (39) becomes
(40)$$V_{\mathrm{NHPF}}=-k_{2}V_{\mathrm{NBPF}}+V_{\mathrm{ILPF}}+k_{2}V_{\mathrm{in}}$$

According to Equations (37), (39), and (40), the second proposed VM second-order multifunction filter system structure can be synthesized. The output signals of NBPF and ILPF, and the second filter parameters of ω_o_ and Q can be derived as follows.
(41)$$\frac{V_{\mathrm{NBPF}}}{V_{\mathrm{in}}}=\frac{s\frac{k_{2}}{\tau_{3}}}{s^{2}+s\frac{k_{2}}{\tau_{3}}+\frac{1}{\tau_{3}\tau_{4}}}$$
(42)$$\frac{V_{\mathrm{ILPF}}}{V_{\mathrm{in}}}=\frac{-k_{2}\frac{1}{\tau_{3}\tau_{4}}}{s^{2}+s\frac{k_{2}}{\tau_{3}}+\frac{1}{\tau_{3}\tau_{4}}}$$
(43)$$\omega_{o}=\sqrt{\frac{1}{\tau_{3}\tau_{4}}}$$
(44)$$Q=\frac{1}{k_{2}}\sqrt{\frac{\tau_{3}}{\tau_{4}}}$$

Equations (43) and (44) indicate that the parameter Q can be independently tuned by adjusting the voltage gain building block of k_2_ without affecting the parameter ω_o_. In the special case of τ_3_ = τ_4_ = τ, the parameter ω_o_ can also be independently tuned by τ without affecting the parameter Q. Based on the DC bias current I_B_ of the corresponding LT1228, the second proposed filter parameters of ω_o_ and Q can be controlled electronically and orthogonally.

In Equation (34), the VM NHPF second-order transfer function is decomposed into the VM NBPF and ILPF second-order transfer functions, and the synthesis of the system block diagram can be achieved. Equations (37), (39), and (40) can be rearranged as Equation (45) in the form of an input–output matrix equation, and the system block diagram synthesis for the second proposed VM LT1228-based second-order multifunction filter design is shown in Figure 5. It uses one input voltage node and three output voltage nodes, and consists of a non-inverting lossless integrator, an inverting lossless integrator, and a voltage gain building block.
(45)$$\left[\begin{array}{ccc} 1 & 0 & -\frac{1}{{s\tau }_{3}}\\ \frac{1}{{s\tau }_{4}} & 1 & 0\\ k_{2} & -1 & 1 \end{array}\right]\left[\begin{array}{c} V_{\mathrm{NBPF}}\\ V_{\mathrm{ILPF}}\\ V_{\mathrm{NHPF}} \end{array}\right]=\left[\begin{array}{c} 0\\ 0\\ k_{2}V_{\mathrm{in}} \end{array}\right]$$

In Figure 5, taking the system blocks of τ_3_ = C_3_/g_m4_, τ_4_ = C_4_/g_m5_, and k_2_ = g_m6_R, Equation (45) can be rewritten as
(46)$$\left[\begin{array}{ccc} 1 & 0 & -\frac{g_{m4}}{{\mathrm{sC}}_{3}}\\ \frac{g_{m5}}{{\mathrm{sC}}_{4}} & 1 & 0\\ g_{m6}R & -1 & 1 \end{array}\right]\left[\begin{array}{c} V_{\mathrm{NBPF}}\\ V_{\mathrm{ILPF}}\\ V_{\mathrm{NHPF}} \end{array}\right]=\left[\begin{array}{c} 0\\ 0\\ g_{m6}{\mathrm{RV}}_{\mathrm{in}} \end{array}\right]$$
where C_3_ and C_4_ are two capacitors, g_m4_, g_m5_, and g_m6_ are three LT1228 transconductance amplifiers, and R is a resistor. Solving Equation (46), the parameters ω_o_ and Q of the second filter and the three VM second-order transfer functions of the NBPF, ILPF, and NHPF can be obtained as follows.
(47)$$\frac{V_{\mathrm{NBPF}}}{V_{\mathrm{in}}}=\frac{{\mathrm{sC}}_{4}g_{m4}g_{m6}R}{s^{2}C_{3}C_{4}+{\mathrm{sC}}_{4}g_{m4}g_{m6}R+g_{m4}g_{m5}}$$
(48)$$\frac{V_{\mathrm{ILPF}}}{V_{\mathrm{in}}}=\frac{-g_{m4}g_{m5}g_{m6}R}{s^{2}C_{3}C_{4}+{\mathrm{sC}}_{4}g_{m4}g_{m6}R+g_{m4}g_{m5}}$$
(49)$$\frac{V_{\mathrm{NHPF}}}{V_{\mathrm{in}}}=\frac{s^{2}C_{3}C_{4}g_{m6}R}{s^{2}C_{3}C_{4}+{\mathrm{sC}}_{4}g_{m4}g_{m6}R+g_{m4}g_{m5}}$$
(50)$$\omega_{o}=\sqrt{\frac{g_{m4}g_{m5}}{C_{3}C_{4}}}$$
(51)$$Q=\frac{1}{g_{m6}R}\sqrt{\frac{C_{3}g_{m5}}{C_{4}g_{m4}}}$$

Equations (50) and (51) indicate that the parameter Q can be independently tuned by adjusting g_m6_ and/or R without affecting the parameter ω_o_. In the special case of C_3_ = C_4_, and g_m4_ = g_m5_ = g_m_, the parameter ω_o_ can also be independently tuned by g_m_ without affecting the parameter Q. Based on the DC bias current I_B_ of the corresponding LT1228, the filter parameters of ω_o_ and Q can be controlled electronically and orthogonally. Based on Figure 5 and Equation (46), Section 2.4 discusses the second proposed high-input impedance electronically tunable VM one-input five-output second-order multifunction filter based on three LT1228 ICs.

### 2.4. Second Proposed VM LT1228-Based Second-Order Multifunction Filter

Based on the synthesis of the system block diagram in Figure 5, the second proposed VM LT1228-based second-order multifunction filter configuration using three commercial LT1228s, five resistors and two capacitors connected to the ground is shown in Figure 6. The nodal analysis of the second proposed VM LT1228-based second-order multifunction filter configuration can be written as follows.
(52)$$V_{o1}=\frac{g_{m4}}{{\mathrm{sC}}_{3}}V_{o3}$$
(53)$$V_{o2}=-\frac{g_{m5}}{{\mathrm{sC}}_{4}}V_{o1}$$
(54)$$V_{o3}=g_{m3}{R(V}_{\mathrm{in}}-V_{o1})+V_{o2}$$
(55)$$V_{o4}=(1+\frac{R_{5}}{R_{6}}{)V}_{o1}$$
(56)$$V_{o5}=(1+\frac{R_{7}}{R_{8}}{)V}_{o2}$$

Equations (52)–(54) can be rearranged in the form of an input–output matrix equation as follows.
(57)$$\left[\begin{array}{ccc} 1 & 0 & -\frac{g_{m4}}{{\mathrm{sC}}_{3}}\\ \frac{g_{m5}}{{\mathrm{sC}}_{4}} & 1 & 0\\ g_{m6}R & -1 & 1 \end{array}\right]\left[\begin{array}{c} V_{o1}\\ V_{o2}\\ V_{o3} \end{array}\right]=\left[\begin{array}{c} 0\\ 0\\ g_{m6}{\mathrm{RV}}_{\mathrm{in}} \end{array}\right]$$

Equation (57) is the same as Equation (46), if we let the outputs V_o1_ = V_NBPF_, V_o2_ = V_ILPF_, and V_o3_ = V_NHPF_. Therefore, the second proposed VM LT1228-based second-order multifunction filter proposed in Figure 6 simultaneously provides the NBPF, ILPF, and NHPF transfer functions at the V_o1_, V_o2_, and V_o3_ outputs, respectively, as shown in Equations (58)–(60).
(58)$$\frac{V_{o1}}{V_{\mathrm{in}}}=\frac{{\mathrm{sC}}_{4}g_{m4}g_{m6}R}{s^{2}C_{3}C_{4}+{\mathrm{sC}}_{4}g_{m4}g_{m6}R+g_{m4}g_{m5}}$$
(59)$$\frac{V_{o2}}{V_{\mathrm{in}}}=\frac{-g_{m4}g_{m5}g_{m6}R}{s^{2}C_{3}C_{4}+{\mathrm{sC}}_{4}g_{m4}g_{m6}R+g_{m4}g_{m5}}$$
(60)$$\frac{V_{o3}}{V_{\mathrm{in}}}=\frac{s^{2}C_{3}C_{4}g_{m6}R}{s^{2}C_{3}C_{4}+{\mathrm{sC}}_{4}g_{m4}g_{m6}R+g_{m4}g_{m5}}$$

Based on Equations (58)–(60), the second proposed filter parameters of ω_o_ and Q are the same as Equations (50) and (51). According to Equations (55) and (56), the second proposed VM LT1228-based second-order multifunction filter has two independent gain-controlled BPF and LPF transfer functions at V_o4_ and V_o5_ output terminals, respectively, as shown in Equations (61) and (62).
(61)$$\frac{V_{o4}}{V_{\mathrm{in}}}=(1+\frac{R_{5}}{R_{6}})(\frac{{\mathrm{sC}}_{4}g_{m4}g_{m6}R}{s^{2}C_{3}C_{4}+{\mathrm{sC}}_{4}g_{m4}g_{m6}R+g_{m4}g_{m5}})$$
(62)$$\frac{V_{o5}}{V_{\mathrm{in}}}=(1+\frac{R_{7}}{R_{8}})(\frac{-g_{m4}g_{m5}g_{m6}R}{s^{2}C_{3}C_{4}+{\mathrm{sC}}_{4}g_{m4}g_{m6}R+g_{m4}g_{m5}})$$

In Equations (61) and (62), the second filter has two independent gain-controlled BPF and LPF transfer functions at the V_o4_ and V_o5_ outputs, and its four resistors of R_5_, R_6_, R_7_, and R_8_ can be tuned to independent gain control without affecting the design parameters of ω_o_ and Q. In Figure 6, the three output voltage nodes V_o3_, V_o4_, and V_o5_ are connected to the corresponding W terminal of each LT1228 to provide low output impedance. If C = C_3_ = C_4_, g_m4_ = g_m5_ = 10I_B4_, and g_m6_ = 10I_B6_, the parameters ω_o_ and Q in Equations (50) and (51) can be rewritten as
(63)$$\omega_{o}=\frac{{10I}_{B4}}{C}$$
(64)$$Q=\frac{1}{{10I}_{B6}R}$$

In this particular case, the parameter ω_o_ can be tuned electronically and independently by the bias current I_B4_ of LT1228 without affecting the parameter Q, and the parameter Q can also be tuned electronically and independently by the bias current I_B6_ of LT1228 without affecting the parameter ω_o_.

## 3. Simulation and Experimental Results

To verify the operation of the two proposed filter topologies in Figure 4 and Figure 6, OrCAD PSpice simulations and experimental verifications were performed based on three commercially available LT1228 ICs. Figure 7 and Figure 8 show the top and bottom views of the printed circuit board (PCB) hardware implementation of the two proposed VM LT1228-based second-order multifunction filters in Figure 4 and Figure 6, respectively. The supply voltage for each circuit is ±15 V. Each power dissipation (PD) calculated in Figure 7 and Figure 8 using the Keithley 2231A-30-3 power supply is 0.69 W. Figure 9 and Figure 10 illustrate the experimental hardware setup used to verify the filter topologies designed in Figure 4 and Figure 6, respectively.

### 3.1. Verification of the First Proposed VM LT1228-Based Multifunction Filter

To verify the operability of the first proposed VM LT1228-based multifunction filter at f_o_ = 159.15 kHz and Q = 1, two capacitors C_1_ = C_2_ = 1 nF, five resistors R = R_1_ = R_2_ = R_3_ = R_4_ = 1 kΩ, and three commercial LT1228 ICs with bias currents of I_B1_ = I_B2_ = I_B3_ = 100 μA were chosen. Figure 11, Figure 12, Figure 13, Figure 14 and Figure 15 illustrate the simulation results of the filter magnitude and phase responses of the first proposed circuit in Figure 4. From Figure 11, Figure 12, Figure 13, Figure 14 and Figure 15, the first proposed VM LT1228-based multifunction filter provides five output responses in the frequency-domain. Figure 16 shows that the simulation results of the Q-value can specify the recommended independent control without affecting the pole frequency. In Figure 16, the simulated Q-value was varied for Q = {2.56, 4.06, 5.01, 5.95} via the three bias currents I_B1_ = I_B2_ = 100 μA and I_B3_ = {40, 25, 20, 16.6} μA. This is expected since the output swing distortion limits the Q-value below 6, and the Q-value error remains below 2%. Figure 17 shows the frequency tunability results of the first proposed filter simulated at the V_o1_ output voltage. In Figure 17, the simulated pole frequency was varied for f_o_ = {46.77, 125.02, 235.5, 961.61} kHz via the three bias currents I_B1_ = I_B2_ = {30, 80, 150, 600} μA and I_B3_ = 100 μA. Figure 16 and Figure 17 confirm that the first VM multifunction filter provides electronic control of the f_o_ and Q parameters. Figure 18 and Figure 19 show the frequency tunability results of the first proposed filter simulated at the V_o2_ and V_o3_ output voltages, respectively. In Figure 18, the simulated pole frequency was varied for f_o_ = {46.61, 124.25, 387.61, 927.25} kHz via the three bias currents I_B1_ = I_B2_ = {30, 80, 250, 600} μA and I_B3_ = 100 μA. In Figure 19, the simulated pole frequency was varied for f_o_ = {46.63, 124.39, 389.22, 936.26} kHz via the three bias currents I_B1_ = I_B2_ = {30, 80, 250, 600} μA and I_B3_ = 100 μA. To demonstrate the stability of the first proposed multifunction filter at different temperatures, the NBPF (V_o1_), NLPF (V_o2_), and IHPF (V_o3_) of the first circuit are operated at different temperatures. Figure 20, Figure 21 and Figure 22 show the NBPF, NLPF, and IHPF operating at different temperatures, respectively. From Figure 20, Figure 21 and Figure 22 it can be seen that the temperature varies from −5° to 50°. The simulated NBPF varies from 174.18 kHz to 144.54 kHz, which affects the operating pole frequency of 159.15 kHz in the range of 9.44% to −9.17%. The simulated NLPF varies from 173.78 kHz to 144.21 kHz, which affects the operating pole frequency of 159.15 kHz in the range 9.19% to −9.38%. The simulated IHPF varies from 174.18 kHz to 144.54 kHz, which affects the operating pole frequency of 159.15 kHz in the range 9.44% to −9.17%. Figure 23, Figure 24, Figure 25, Figure 26 and Figure 27 illustrate the time-domain characteristics of the first proposed circuit when the frequency and amplitude of the input sine wave are 159.15 kHz and 180 m V_pp_, respectively. Table 2 shows the measured phase error between the output and input waveforms at an operating pole frequency of 159.15 kHz. In Table 2, the maximum phase error measured in the time-domain at an operating pole frequency of 159.15 kHz is less than 0.97°. Figure 28 shows the NBPF total harmonic distortion (THD) measured at V_o1_ versus different input voltage signals. In Figure 28, the THD is below 2% when the input peak-to-peak voltage increases to 220 m V_pp_. Figure 29 illustrates the simulated noise performance at the V_o1_ output voltage of the first proposed circuit. As shown in Figure 29, the total equivalent input and output noise voltages at the operating pole frequency were 86.33 and 86.44 nV/$\sqrt{\mathrm{Hz}}$, respectively. Figure 30, Figure 31, Figure 32, Figure 33 and Figure 34 show the first filter magnitude and phase responses measured by the Keysight E5061B-3L5 network analyzer for the first proposed circuit in Figure 4. Figure 35, Figure 36, Figure 37, Figure 38 and Figure 39 illustrate the calculated, simulated, and measured filter amplitude and phase responses for the first proposed circuit in Figure 4. Based on the ideal pole phase marked in the frequency-domain characteristics, Table 3 shows the simulated and measured pole frequency errors for the first proposed circuit. In Table 3, the maximum percentage error of the pole frequency measured in the frequency-domain is less than 1.22%. According to Figure 35, Figure 36, Figure 37, Figure 38 and Figure 39 and Table 3, the amplitude and phase responses of the first proposed circuit agree with the simulated and measured results.

In both simulations and measurements, the bias current I_B_ of commercial LT1228 IC was tuned in the range of 30 to 600 μA, making the first filter’s pole frequency f_o_ easily adjustable in the range of 47.74 to 954.92 kHz. Figure 40 and Figure 41 show the frequency tunability results of the first proposed filter measured at the V_o1_ output voltage. In Figure 40, the measured pole frequency varied for f_o_ = {94.92, 189.21, 349.03, 635.39} kHz via the three bias currents I_B1_ = I_B2_ = {60, 120, 220, 400} μA and I_B3_ = 100 μA. In Figure 41, the measured pole frequency varied for f_o_ = {47.2, 126.56, 236.18, 951.94} kHz via the three bias currents I_B1_ = I_B2_ = {30, 80, 150, 600} μA and I_B3_ = 100 μA. Figure 42 and Figure 43 illustrate the calculated, simulated, and measured filter amplitude responses at the V_o1_ output voltage. As shown in Figure 42 and Figure 43, the electronic tunability of the first filter parameter f_o_ does not affect the parameter Q. Figure 44 shows the quality factor tunability results for the first proposed filter at the V_o1_ output voltage. In Figure 44, the measured quality factor varied for Q = {0.92, 1.56, 1.97, 2.79} via the three bias currents I_B1_ = I_B2_ = 100 μA and I_B3_ = {142, 66.6, 50, 33.3} μA. Figure 45 illustrates the calculated, simulated, and measured amplitude responses of the first filter at the V_o1_ output voltage. As shown in Figure 45, the electronic tunability of the first filter parameter Q does not affect the parameter f_o_. To show the linearity of the first proposed VM LT1228-based multifunction filter in Figure 4, a 1-dB power gain compression point (P1dB) of the NBPF was measured at the V_o1_ output voltage. Figure 46 shows that the measured input P1dB point is approximately −7.1 dBm. Figure 47 shows the spectrum analysis measured at the V_o1_ output voltage of the first proposed circuit when the frequency and amplitude of the input sine wave are 159.15 kHz and 180 m V_pp_, respectively. As shown in Figure 47, the spurious-free dynamic range (SFDR) between the first tone and the highest spur of the other levels is 45.02 dBc. To show the non-linearity of the first proposed circuit in Figure 4, a two-tone test has been performed. The intermodulation distortion (IMD) performance of the first proposed circuit at the V_o1_ output voltage is further investigated using equal-amplitude two-tone signals with frequencies f_1_ = 158.15 kHz and f_2_ = 160.15 kHz. Using the Keysight-Agilent N9000A CXA signal analyzer, Figure 48 illustrates the IMD results measured at the V_o1_ output voltage of the first proposed circuit. In Figure 48, the third-order IMD (IMD3) and third-order intercept (TOI) point were measured as −48.84 dBc and 4.133 dBm, respectively. Table 4 summarizes the measured performance of the first proposed VM LT1228-based second-order multifunction filter.

### 3.2. Verification of the Second Proposed VM LT1228-Based Multifunction Filter

To verify the operability of the second proposed VM LT1228-based multifunction filter at f_o_ = 159.15 kHz and Q = 1, two capacitors C_3_ = C_4_ = 1 nF, five resistors R = R_5_ = R_6_ = R_7_ = R_8_ = 1 kΩ, and three commercial LT1228 ICs with bias currents of I_B4_ = I_B5_ = I_B6_ = 100 μA were chosen. Figure 49, Figure 50, Figure 51, Figure 52 and Figure 53 illustrate the simulation results of the filter magnitude and phase responses of the second proposed circuit in Figure 6. From Figure 49, Figure 50, Figure 51, Figure 52 and Figure 53, the second proposed VM LT1228-based multifunction filter provides five output responses in the frequency-domain. Figure 54 shows that the simulation results of the Q-value can specify the recommended independent control without affecting the pole frequency. In Figure 54, the simulated Q-value varied for Q = {2.56, 4.06, 5.01, 5.95} via the three bias currents I_B4_ = I_B5_ = 100 μA and I_B6_ = {40, 25, 20, 16.6} μA. This is expected since the output swing distortion limits the Q-value below 6, and the Q-value error remains below 2%. Figure 55 shows the frequency tunability results of the second proposed filter simulated at the V_o1_ output voltage. In Figure 55, the simulated pole frequency was varied for f_o_ = {46.77, 125.02, 235.5, 879.02} kHz via the three bias currents I_B4_ = I_B5_ = {30, 80, 150, 550} μA and I_B6_ = 100 μA. Figure 54 and Figure 55 confirm that the second VM multifunction filter provides electronic control of the f_o_ and Q parameters. Figure 56 and Figure 57 show the frequency tunability results of the second proposed filter simulated at the V_o2_ and V_o3_ output voltages, respectively. In Figure 56, the simulated pole frequency was varied for f_o_ = {46.55, 124.16, 387.25, 849.18} kHz via the three bias currents I_B4_ = I_B5_ = {30, 80, 250, 550} μA and I_B6_ = 100 μA. In Figure 57, the simulated pole frequency was varied for f_o_ = {46.66, 124.45, 389.04, 857.03} kHz via the three bias currents I_B4_ = I_B5_ = {30, 80, 250, 550} μA and I_B6_ = 100 μA. To demonstrate the stability of the second proposed multifunction filter at different temperatures, the NBPF (V_o1_), ILPF (V_o2_), and NHPF (V_o3_) of the second circuit are operated at different temperatures. Figure 58, Figure 59 and Figure 60 show the NBPF, ILPF, and NHPF operating at different temperatures, respectively. From Figure 58, Figure 59 and Figure 60, the temperature varies from −5° to 50°. The simulated NBPF varied from 174.18 kHz to 144.54 kHz, which affects the operating pole frequency of 159.15 kHz in the range of 9.44% to −9.17%. The simulated ILPF varied from 173.78 kHz to 144.21 kHz, which affects the operating pole frequency of 159.15 kHz in the range 9.19% to −9.38%. The simulated NHPF varied from 174.18 kHz to 144.54 kHz, which affects the operating pole frequency of 159.15 kHz in the range 9.44% to −9.17%. Figure 61, Figure 62, Figure 63, Figure 64 and Figure 65 illustrate the calculated, simulated, and measured filter amplitude and phase responses of the second proposed circuit in Figure 6. Based on the ideal pole phase marked in the frequency-domain characteristics, Table 5 shows the simulated and measured pole frequency errors for the second proposed circuit. In Table 5, the maximum percentage error of the pole frequency measured in the frequency-domain is less than 1.55%. According to Figure 61, Figure 62, Figure 63, Figure 64 and Figure 65 and Table 5, the amplitude and phase responses of the second proposed circuit were in agreement with the simulated and measured results. Figure 66 and Figure 67 illustrate the calculated, simulated, and measured amplitude responses of the second filter at the V_o1_ output voltage. In Figure 66, the measured pole frequency was varied for f_o_ = {47.71, 127.01, 237.05, 887.34} kHz via the three bias currents I_B4_ = I_B5_ = {30, 80, 150, 550} μA and I_B6_ = 100 μA. In Figure 67, the measured quality factor varied for Q = {0.88, 1.62, 2.05, 2.9} via the three bias currents I_B4_ = I_B5_ = 100 μA and I_B6_ = {142, 66.6, 50, 33.3} μA. Figure 68, Figure 69, Figure 70, Figure 71 and Figure 72 illustrate the time-domain characteristics of the second proposed circuit when the frequency and amplitude of the input sine wave are 159.15 kHz and 180 m V_pp_, respectively. Table 6 shows the measured phase error between the output and input waveforms at an operating pole frequency of 159.15 kHz. In Table 6, the maximum phase error measured in the time-domain at an operating pole frequency of 159.15 kHz is less than 2.55°. Figure 73 shows the spectrum of the second proposed filter measured at the V_o1_ output voltage. In Figure 73, the frequency and amplitude of the input sine wave were 159.15 kHz and 180 m V_pp_, respectively. As shown in Figure 73, the SFDR between the first tone and the highest spur of the other levels is 45.88 dBc, and the calculated THD value is 0.6%. Figure 74 shows the THD measured at V_o1_ versus different input voltage signals. In Figure 74, the THD is below 2% when the input peak-to-peak voltage increases to 220 m V_pp_. The P1dB performance of the second proposed circuit is measured at the V_o1_ output voltage, and the measured input P1dB point is approximately −7 dBm, as shown in Figure 75. The IMD performance of the second proposed circuit NBPF at the V_o1_ output voltage is investigated using equal-amplitude two-tone signals with frequencies f_1_ = 158.15 kHz and f_2_ = 160.15 kHz. Figure 76 shows the IMD results in Figure 6 for the second proposed circuit NBPF at the V_o1_ output voltage. In Figure 76, the IMD3 and TOI point were measured as −49.65 dBc and 4.316 dBm, respectively. Table 7 summarizes the measured performance of the second proposed VM LT1228-based second-order multifunction filter.

## 4. Conclusions

This paper presents the syntheses of two new VM electronically tunable one-input five-output second-order BPF, LPF, and HPF transfer functions based on three LT1228 ICs. Both configurations with a single input voltage terminal and five output voltage terminals use three commercial LT1228 ICs and seven passive components. These two newly synthesized VM second-order multifunction filters can simultaneously provide the following eight attractive advantages: (i) Both circuits can generate BPF, LPF, and HPF transfer functions simultaneously, making them suitable for use in three-way crossover networks. (ii) Both circuits have one high-impedance input, making them suitable for cascading input voltages. (iii) Both circuits have three low-impedance outputs, making them suitable for cascading three output voltages. (iv) The parameters ω_o_ and Q of the two filters permit electronic and orthogonal tuning. (v) The parameter Q of the two filters has independent and electronic tuning capability. (vi) Passive components do not require matching conditions. (vii) The passband gains of the BPF and LPF responses can be controlled effectively and independently without affecting the filter parameters ω_o_ and Q. (viii) Synthesis method of the electronically tunable VM second-order multifunction filter topologies based on the non-inverting/inverting HPF second-order transfer functions. Circuit design and implementation results were obtained to demonstrate these two VM LT1228-based second-order multifunction filters. The measured input P1dB, IMD3, TOI, and SFDR of the first filter were −7.1 dBm, −48.84 dBc, 4.133 dBm, and 45.02 dBc, respectively. The measured input P1dB, IMD3, TOI, and SFDR of the second filter were −7 dBm, −49.65 dBc, 4.316 dBm, and 45.88 dBc, respectively. Both circuits use a topology synthesis method based on the VM second-order non-inverting/inverting HP filter transfer functions to generate the BP, LP, and HP filter transfer functions simultaneously, making them suitable for three-way crossover network high-fidelity loudspeaker applications. Three commercial LT1228 ICs and seven passive components were used in OrCAD PSpice simulations and experimental measurements to verify the operation of the two proposed VM LT1228-based second-order multifunction filter topologies.

## Figures and Tables

**Figure 1 sensors-22-09379-f001:**
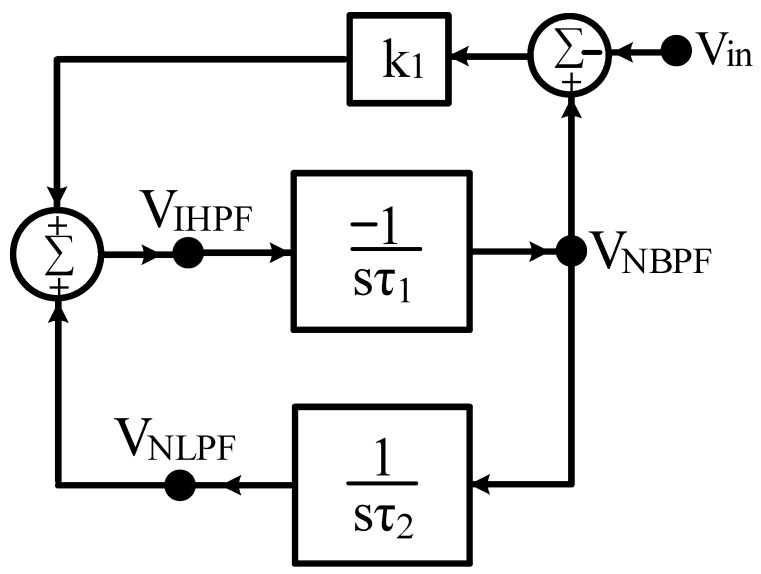
Synthesis of the first proposed VM LT1228-based filter system module with two integrator loops and a voltage gain building block.

**Figure 2 sensors-22-09379-f002:**
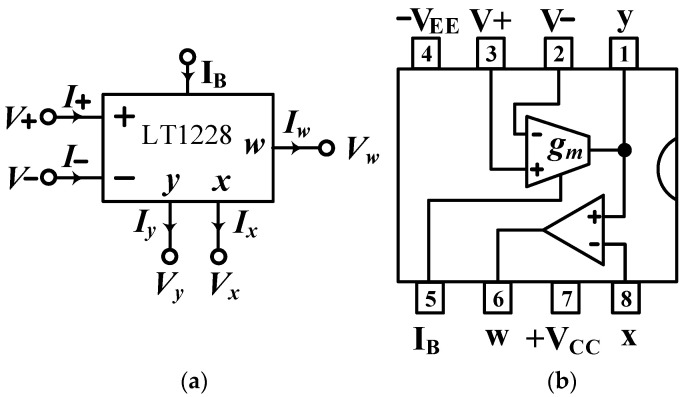
LT1228 (**a**) circuit symbol, and (**b**) IC package in an 8-pin configuration.

**Figure 3 sensors-22-09379-f003:**
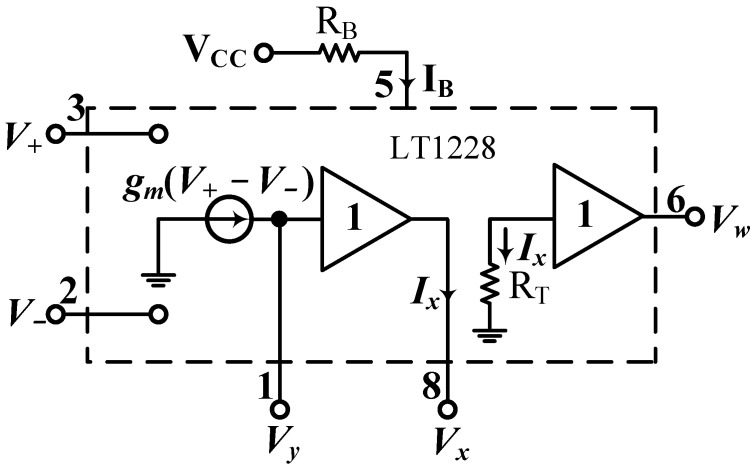
Equivalent circuit of LT1228.

**Figure 4 sensors-22-09379-f004:**
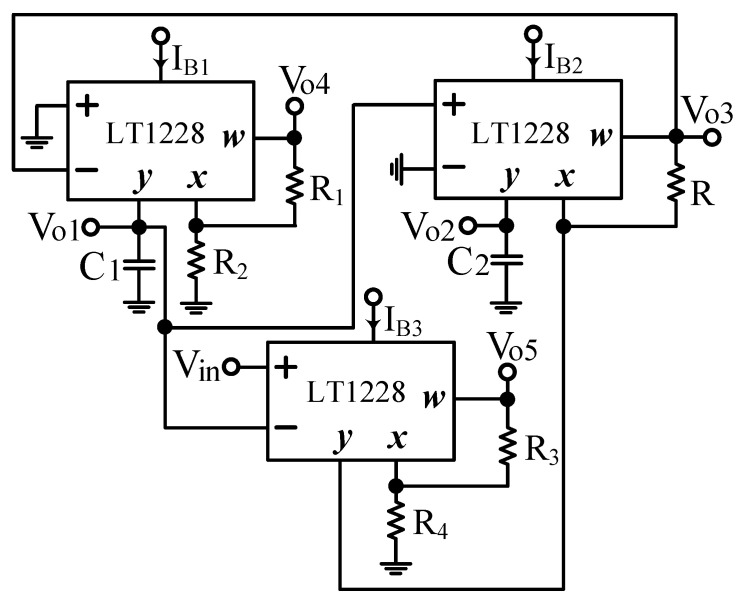
First proposed VM LT1228-based second-order multifunction filter configuration.

**Figure 5 sensors-22-09379-f005:**
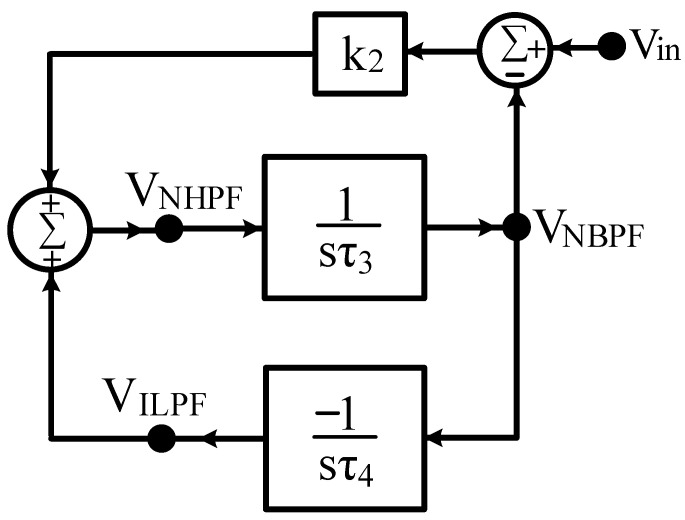
Synthesis of the second proposed VM LT1228-based filter system module with two integrator loops and a voltage gain building block.

**Figure 6 sensors-22-09379-f006:**
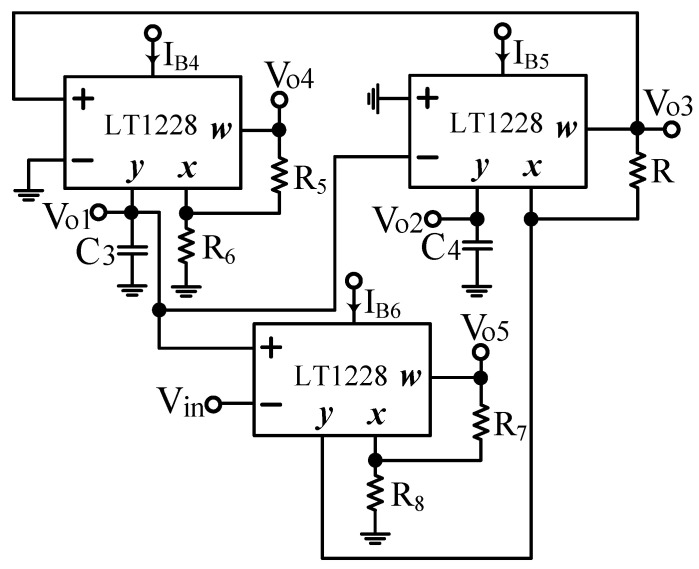
Second proposed VM LT1228-based second-order multifunction filter configuration.

**Figure 7 sensors-22-09379-f007:**
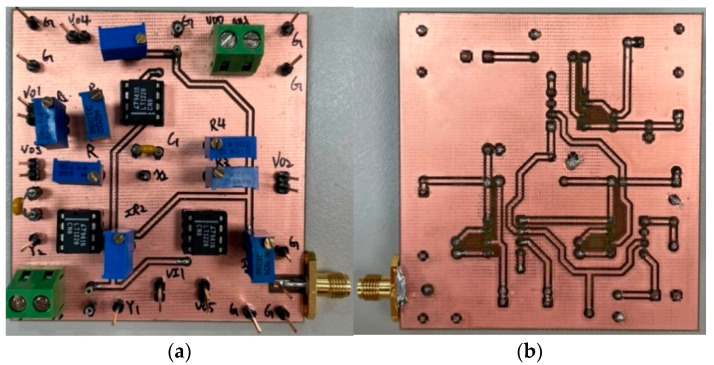
(**a**) Top and (**b**) bottom photos of the first proposed VM LT1228-based multifunction filter PCB hardware implementation in Figure 4.

**Figure 8 sensors-22-09379-f008:**
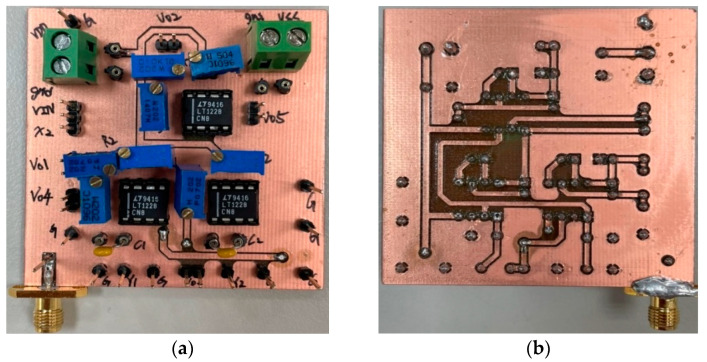
(**a**) Top and (**b**) bottom photos of the second proposed VM LT1228-based multifunction filter PCB hardware implementation in Figure 6.

**Figure 9 sensors-22-09379-f009:**
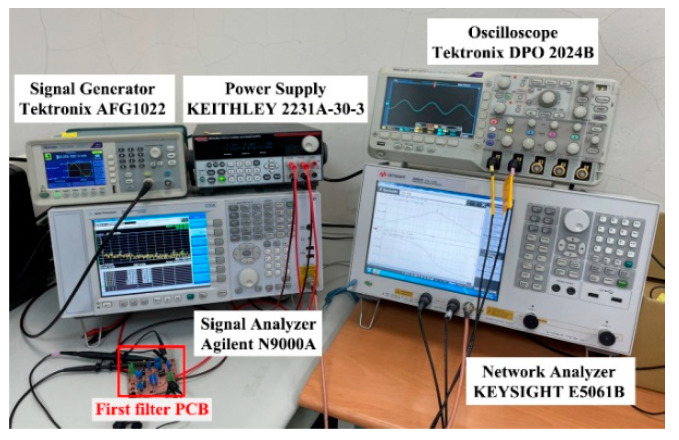
Experimental hardware setup of the first proposed VM LT1228-based multifunction filter in Figure 4.

**Figure 10 sensors-22-09379-f010:**
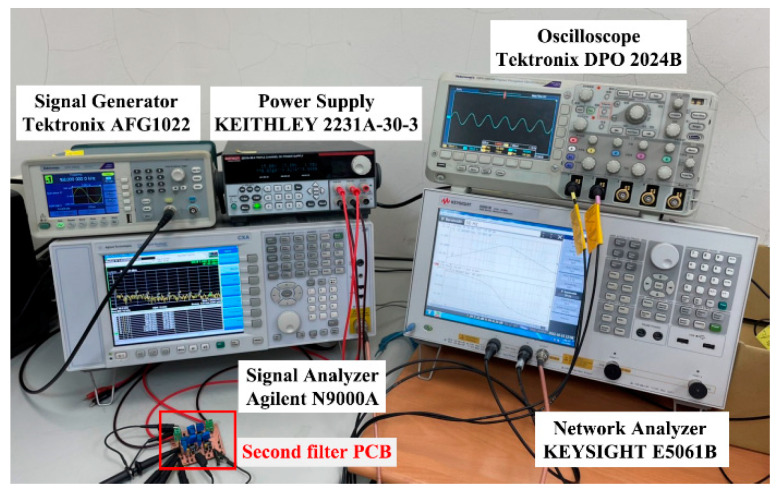
Experimental hardware setup of the second proposed VM LT1228-based multifunction filter in Figure 6.

**Figure 11 sensors-22-09379-f011:**
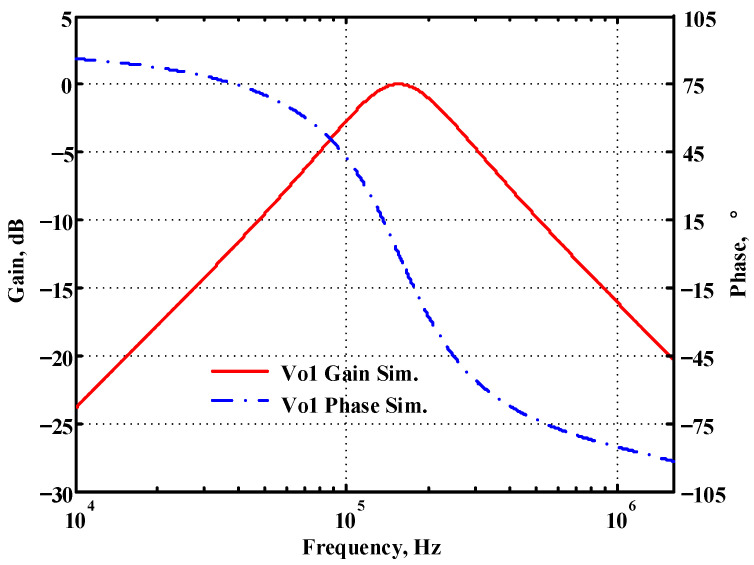
Simulated gain and phase frequency response of the first proposed circuit at V_o1_.

**Figure 12 sensors-22-09379-f012:**
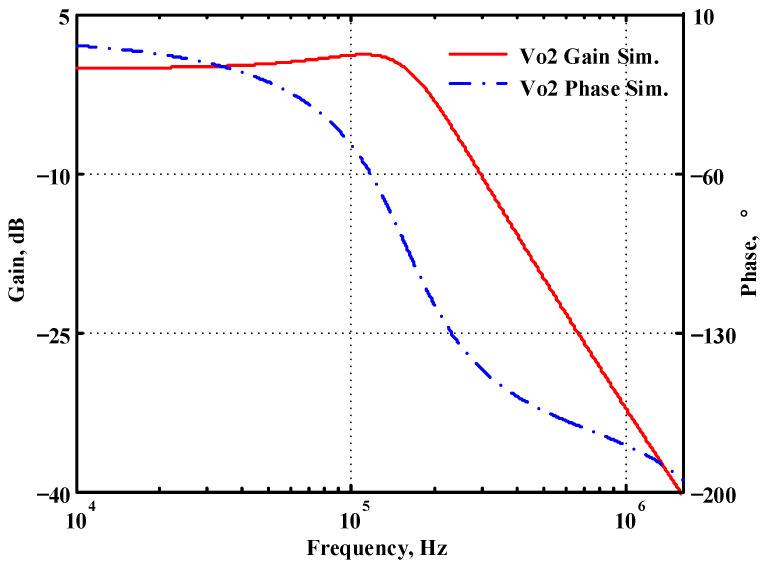
Simulated gain and phase frequency response of the first proposed circuit at V_o2_.

**Figure 13 sensors-22-09379-f013:**
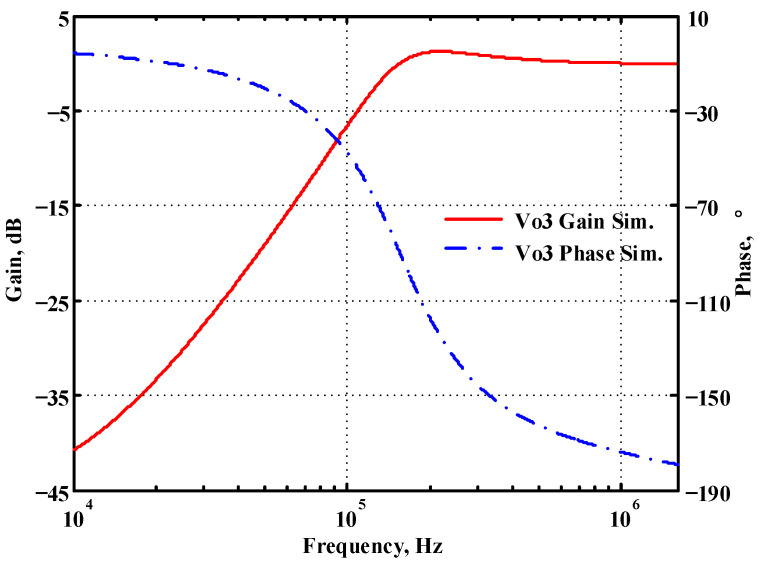
Simulated gain and phase frequency response of the first proposed circuit at V_o3_.

**Figure 14 sensors-22-09379-f014:**
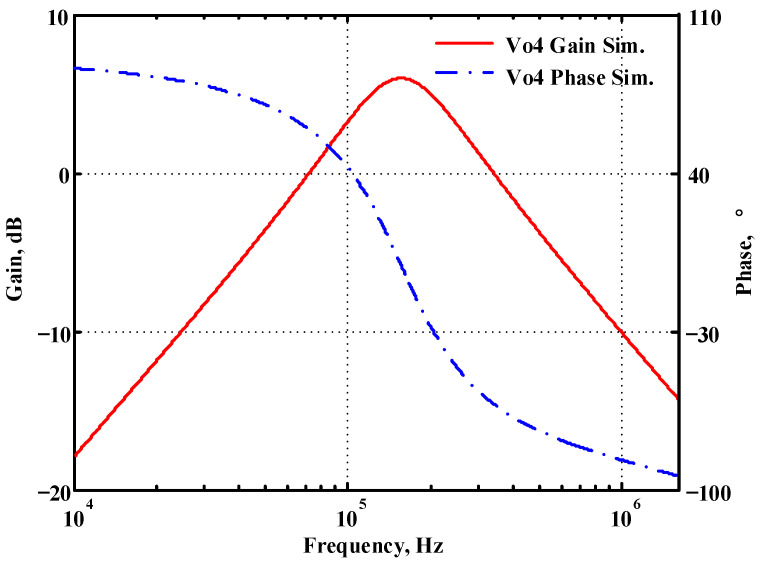
Simulated gain and phase frequency response of the first proposed circuit at V_o4_.

**Figure 15 sensors-22-09379-f015:**
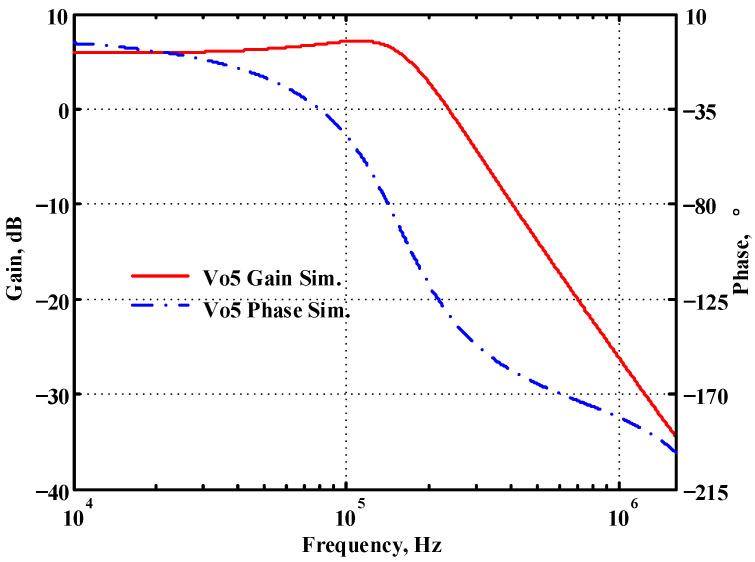
Simulated gain and phase frequency response of the first proposed circuit at V_o5_.

**Figure 16 sensors-22-09379-f016:**
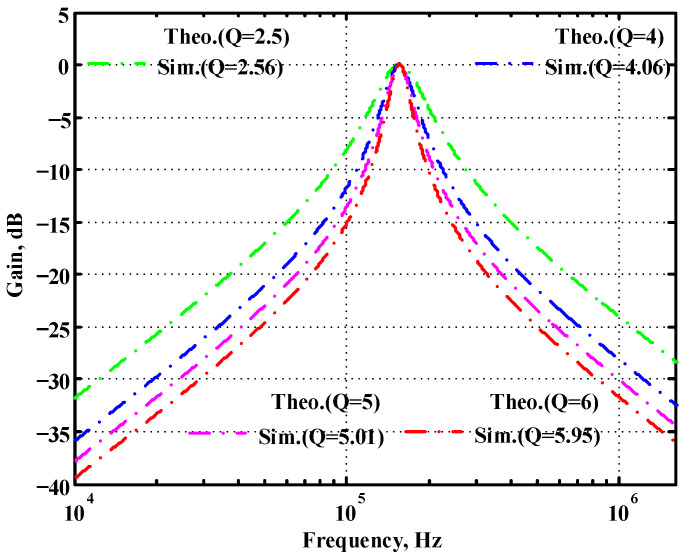
Simulated gain response of the first proposed circuit at V_o1_ by changing Q and keeping f_o_ constant.

**Figure 17 sensors-22-09379-f017:**
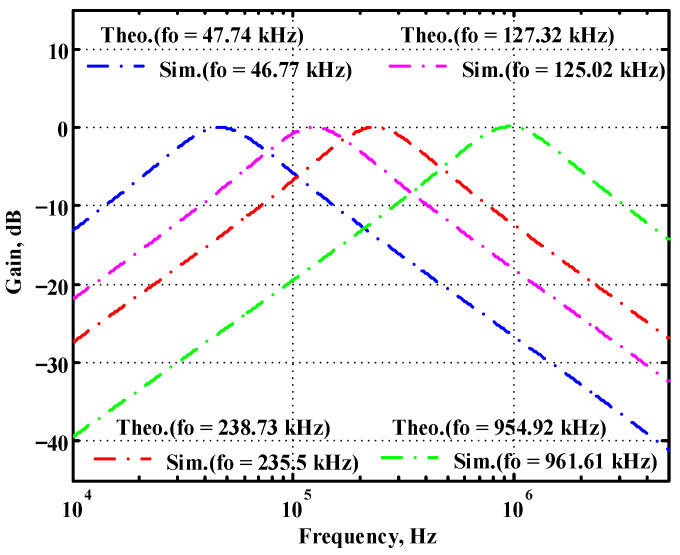
Simulated gain response of the first proposed circuit at V_o1_ by changing f_o_ and keeping Q constant.

**Figure 18 sensors-22-09379-f018:**
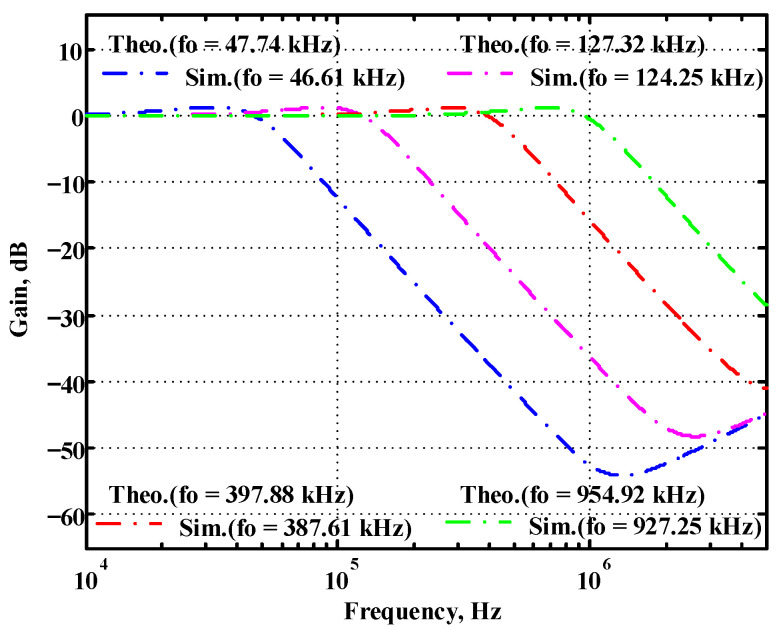
Simulated gain response of the first proposed circuit at V_o2_ by changing f_o_ and keeping Q constant.

**Figure 19 sensors-22-09379-f019:**
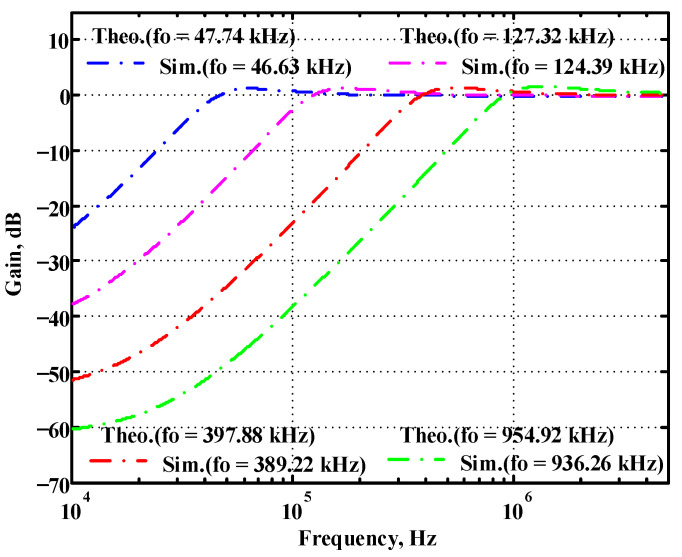
Simulated gain response of the first proposed circuit at V_o3_ by changing f_o_ and keeping Q constant.

**Figure 20 sensors-22-09379-f020:**
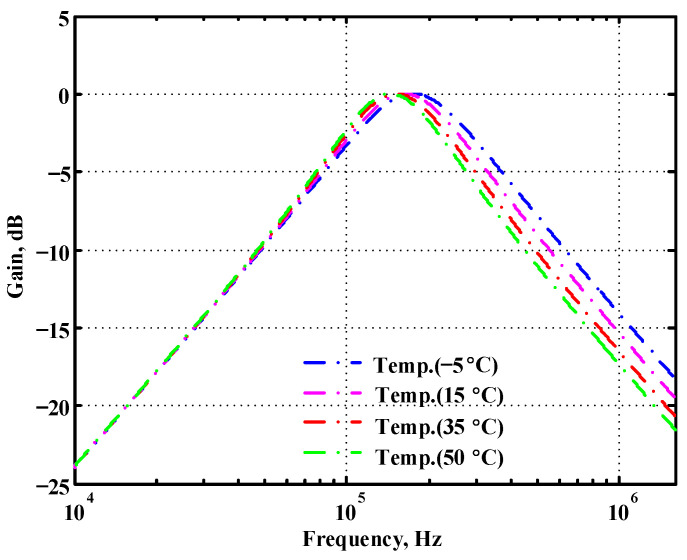
Simulated temperature variation at V_o1_ for the first proposed circuit.

**Figure 21 sensors-22-09379-f021:**
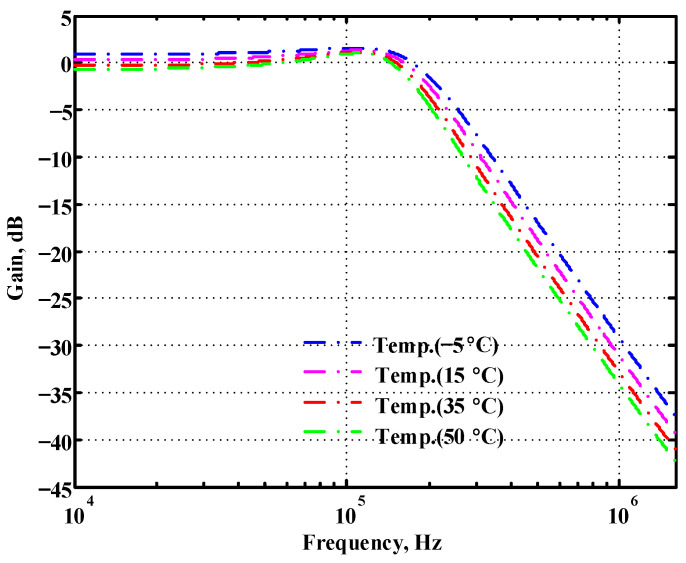
Simulated temperature variation at V_o2_ for the first proposed circuit.

**Figure 22 sensors-22-09379-f022:**
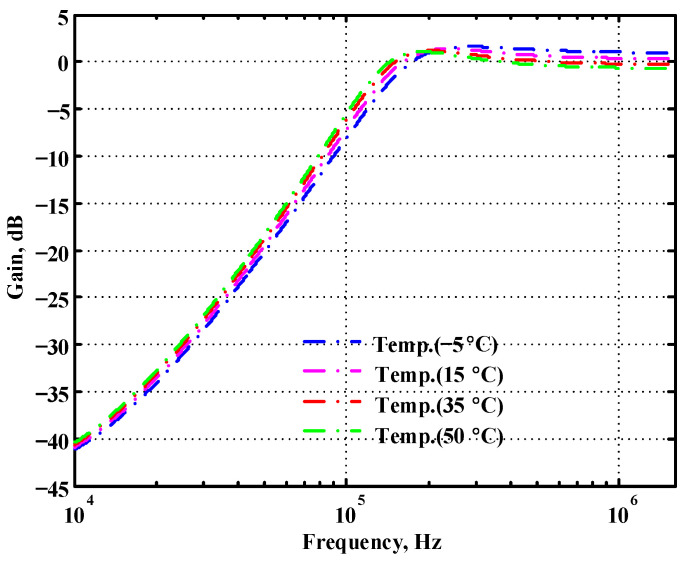
Simulated temperature variation at V_o3_ for the first proposed circuit.

**Figure 23 sensors-22-09379-f023:**
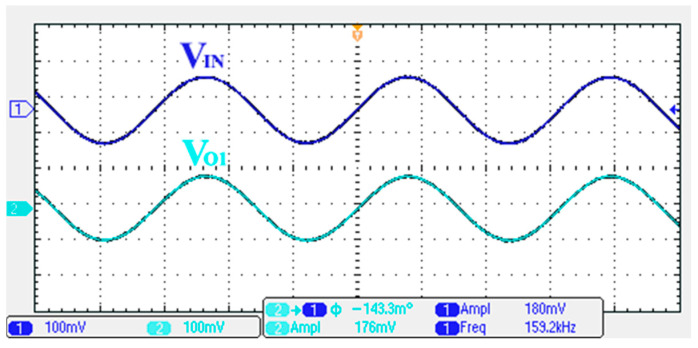
Measured output and input characteristics of the first proposed circuit at V_o1_.

**Figure 24 sensors-22-09379-f024:**
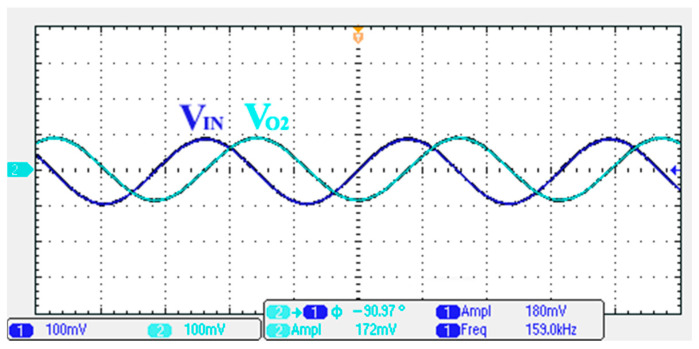
Measured output and input characteristics of the first proposed circuit at V_o2_.

**Figure 25 sensors-22-09379-f025:**
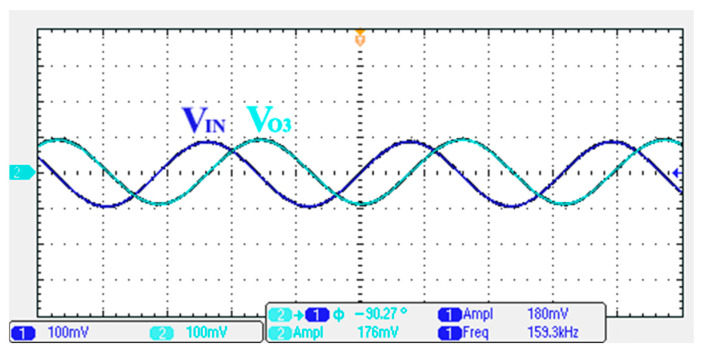
Measured output and input characteristics of the first proposed circuit at V_o3_.

**Figure 26 sensors-22-09379-f026:**
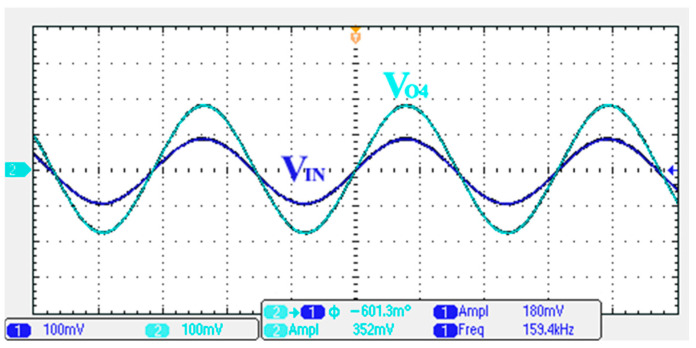
Measured output and input characteristics of the first proposed circuit at V_o4_.

**Figure 27 sensors-22-09379-f027:**
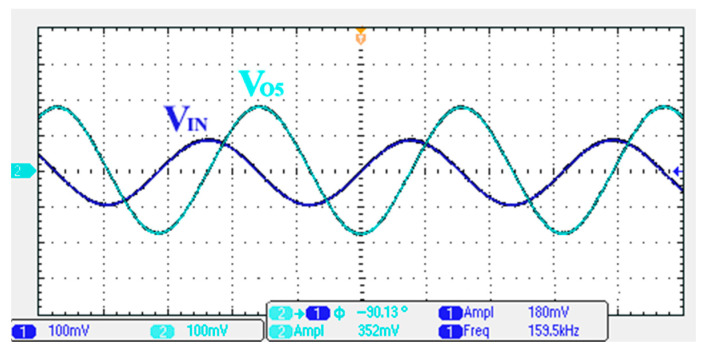
Measured output and input characteristics of the first proposed circuit at V_o5_.

**Figure 28 sensors-22-09379-f028:**
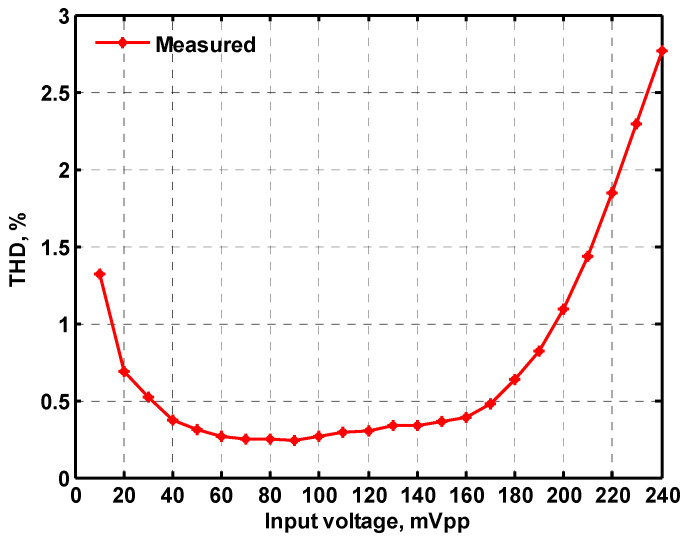
Measured THD of the first proposed circuit at V_o1_ in Figure 4 versus peak-to-peak input voltage signal at 159.15 kHz.

**Figure 29 sensors-22-09379-f029:**
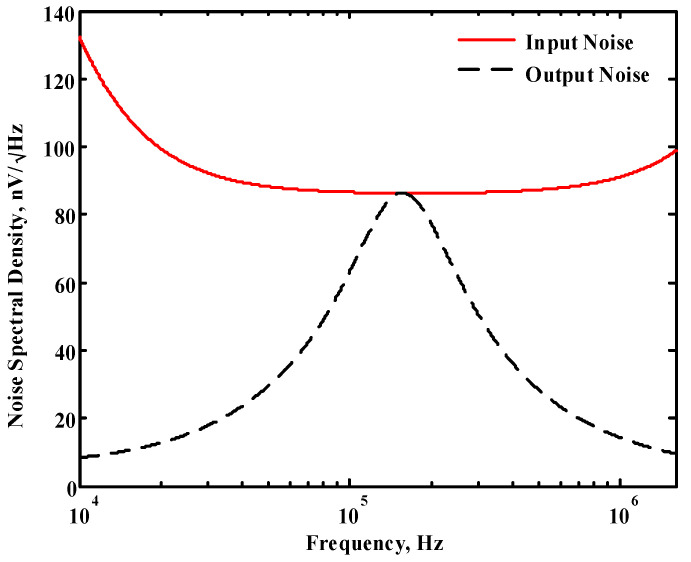
Simulated noise performance of the first proposed circuit at V_o1_.

**Figure 30 sensors-22-09379-f030:**
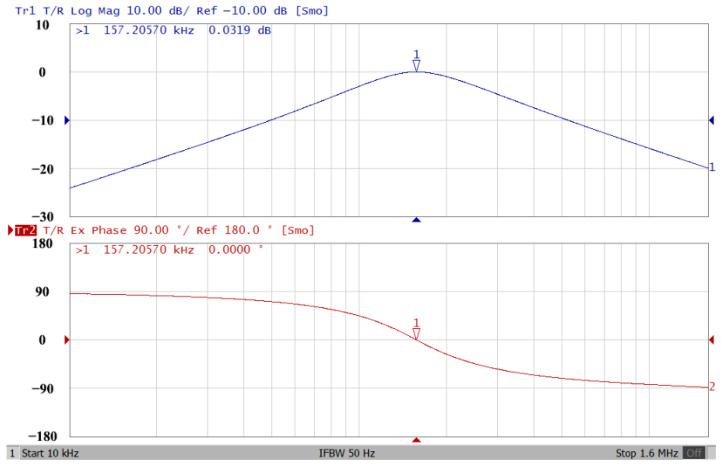
Frequency-domain characteristics of the measured amplitude and phase responses of the first proposed circuit at V_o1_.

**Figure 31 sensors-22-09379-f031:**
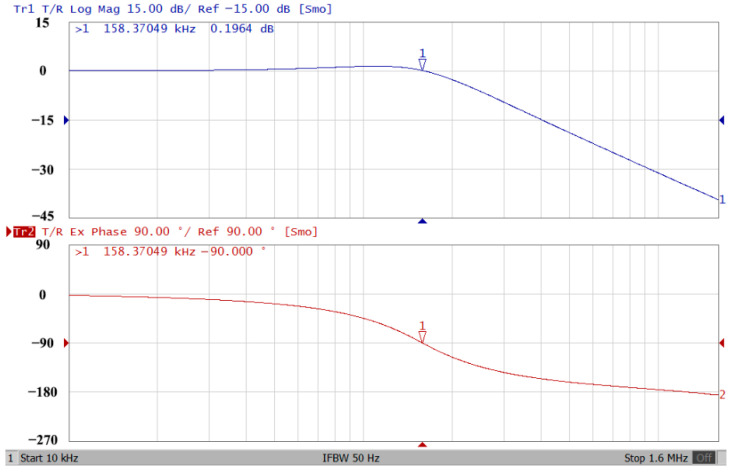
Frequency-domain characteristics of the measured amplitude and phase responses of the first proposed circuit at V_o2_.

**Figure 32 sensors-22-09379-f032:**
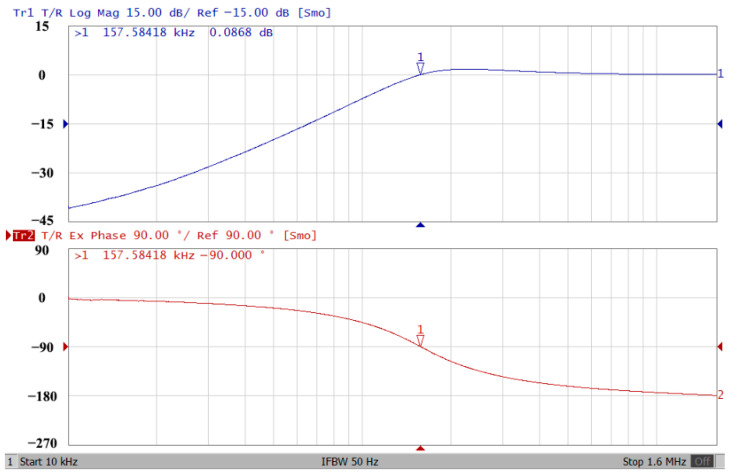
Frequency-domain characteristics of the measured amplitude and phase responses of the first proposed circuit at V_o3_.

**Figure 33 sensors-22-09379-f033:**
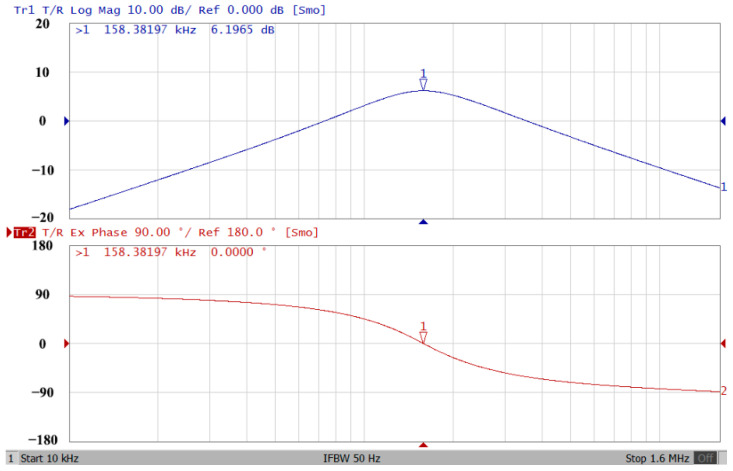
Frequency-domain characteristics of the measured amplitude and phase responses of the first proposed circuit at V_o4_.

**Figure 34 sensors-22-09379-f034:**
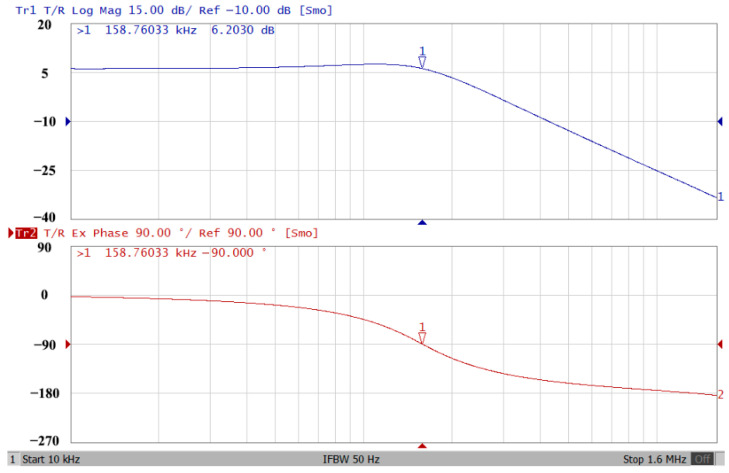
Frequency-domain characteristics of the measured amplitude and phase responses of the first proposed circuit at V_o5_.

**Figure 35 sensors-22-09379-f035:**
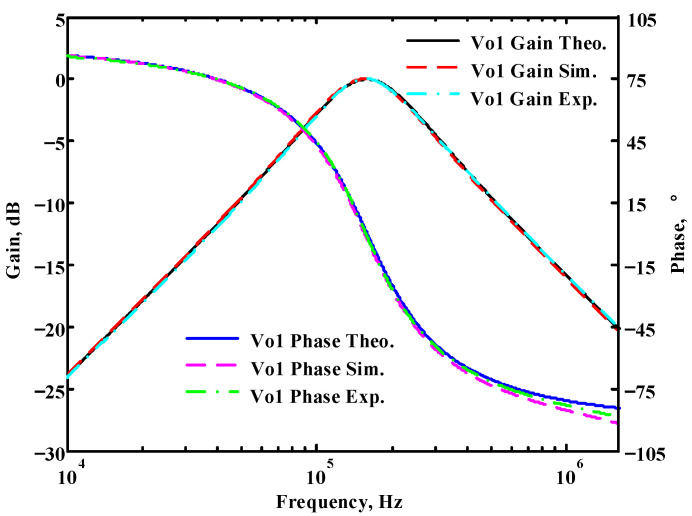
Simulated, measured, and theoretical comparison results of the first proposed circuit at V_o1_.

**Figure 36 sensors-22-09379-f036:**
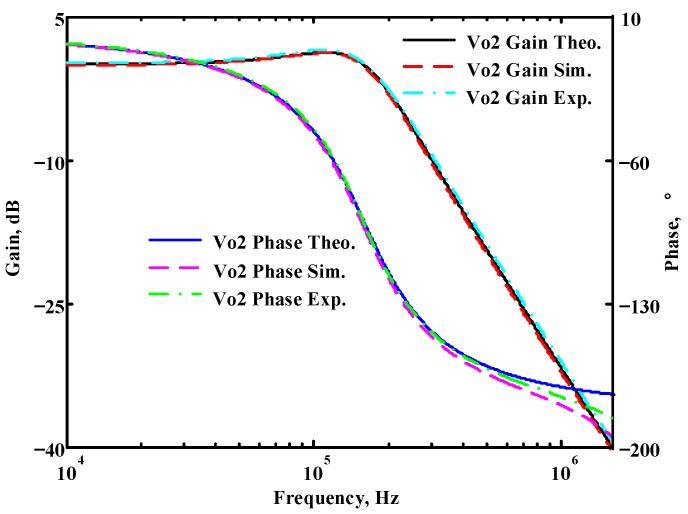
Simulated, measured, and theoretical comparison results of the first proposed circuit at V_o2_.

**Figure 37 sensors-22-09379-f037:**
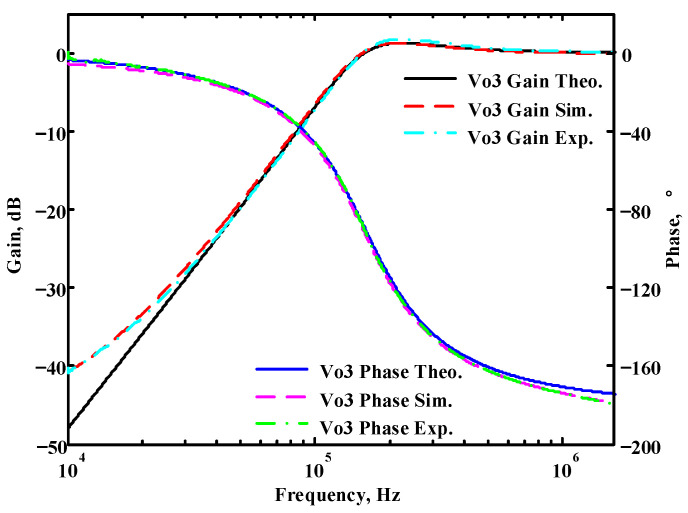
Simulated, measured, and theoretical comparison results of the first proposed circuit at V_o3_.

**Figure 38 sensors-22-09379-f038:**
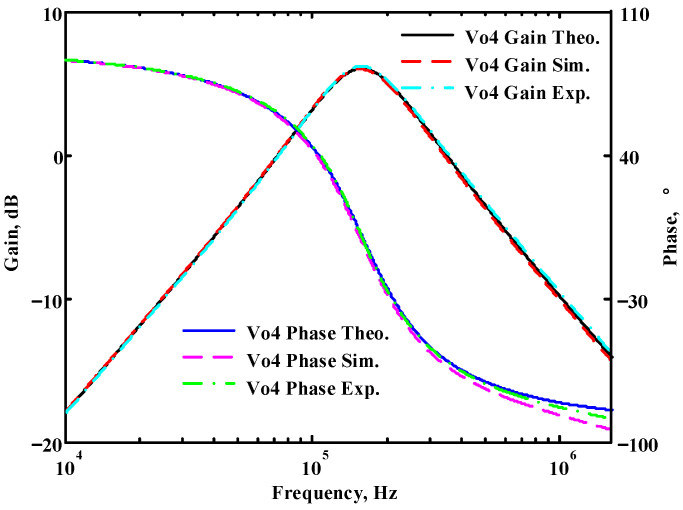
Simulated, measured, and theoretical comparison results of the first proposed circuit at V_o4_.

**Figure 39 sensors-22-09379-f039:**
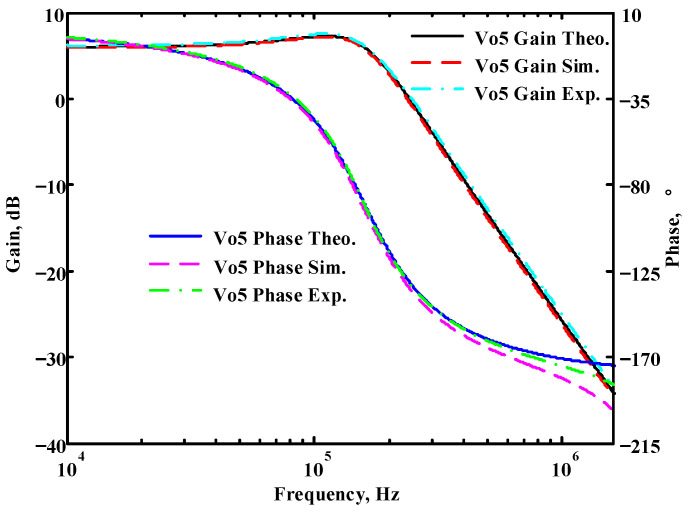
Simulated, measured, and theoretical comparison results of the first proposed circuit at V_o5_.

**Figure 40 sensors-22-09379-f040:**
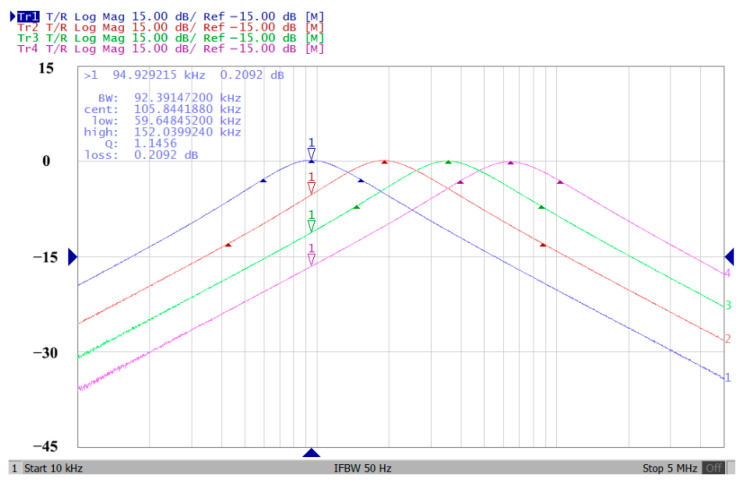
Electronic tunability f_o_ measured at V_o1_ from 94.92 to 635.39 kHz for the first proposed circuit without affecting the Q value.

**Figure 41 sensors-22-09379-f041:**
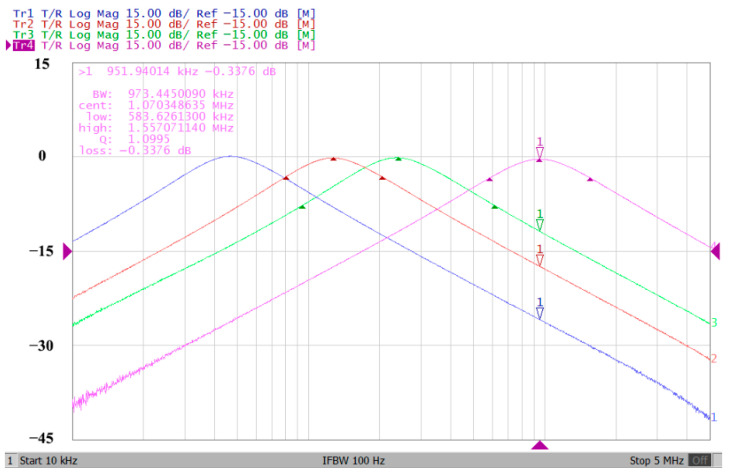
Electronic tunability f_o_ measured at V_o1_ from 47.2 to 951.94 kHz for the first proposed circuit without affecting the Q value.

**Figure 42 sensors-22-09379-f042:**
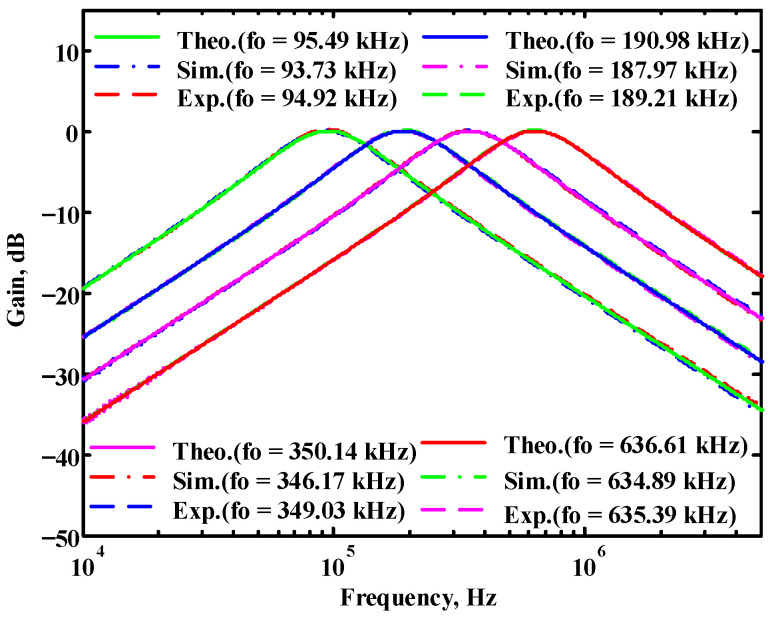
Simulated, measured, and theoretical comparison results of the electronic tunability f_o_ at V_o1_ in Figure 40 without affecting the Q value.

**Figure 43 sensors-22-09379-f043:**
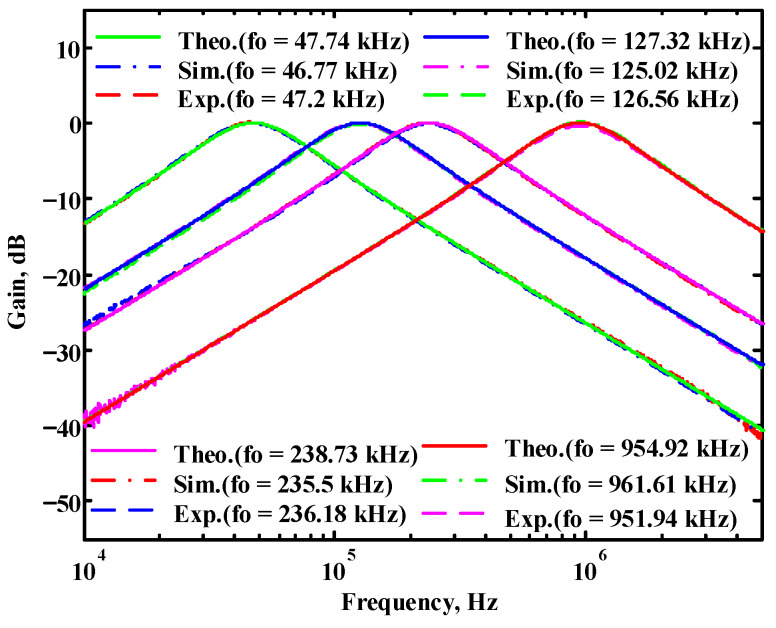
Simulated, measured, and theoretical comparison results of the electronic tenability f_o_ at V_o1_ in Figure 41 without affecting the Q value.

**Figure 44 sensors-22-09379-f044:**
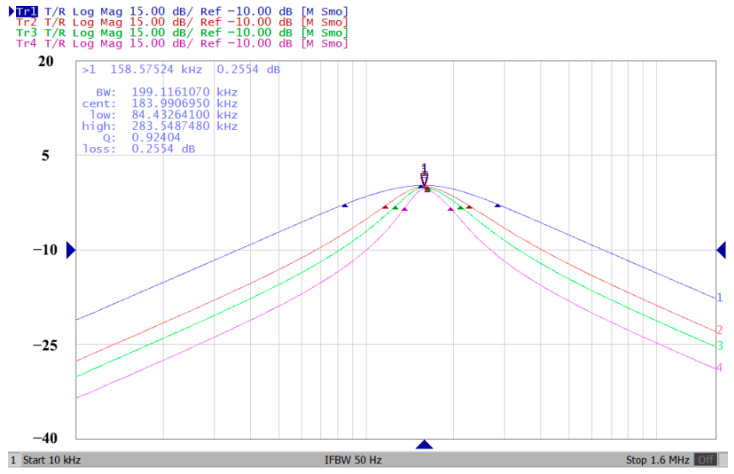
Measured electronic tunability Q of the first proposed circuit at V_o1_ without affecting the f_o_ value.

**Figure 45 sensors-22-09379-f045:**
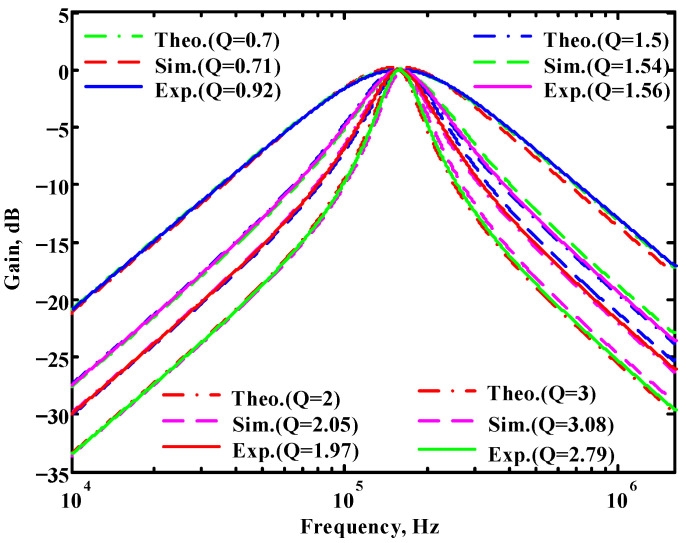
Simulated, measured, and theoretical comparison results of the electronic tunability Q for the first proposed circuit at V_o1_ without affecting the f_o_ value.

**Figure 46 sensors-22-09379-f046:**
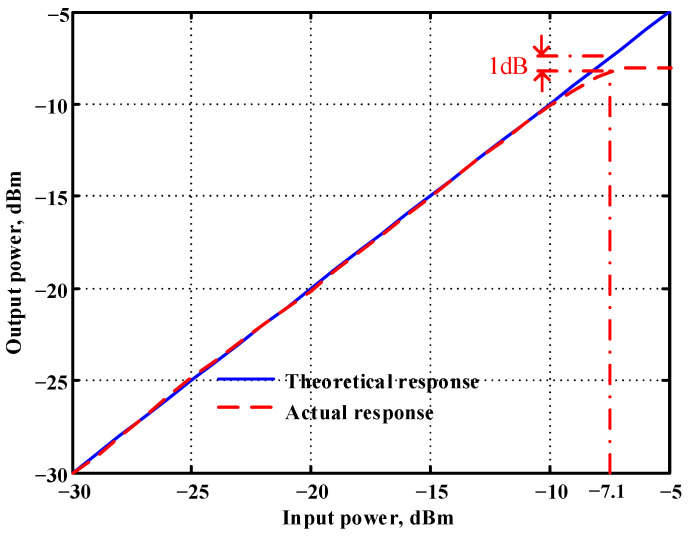
Measured results of the P1dB point of the first proposed circuit at V_o1_.

**Figure 47 sensors-22-09379-f047:**
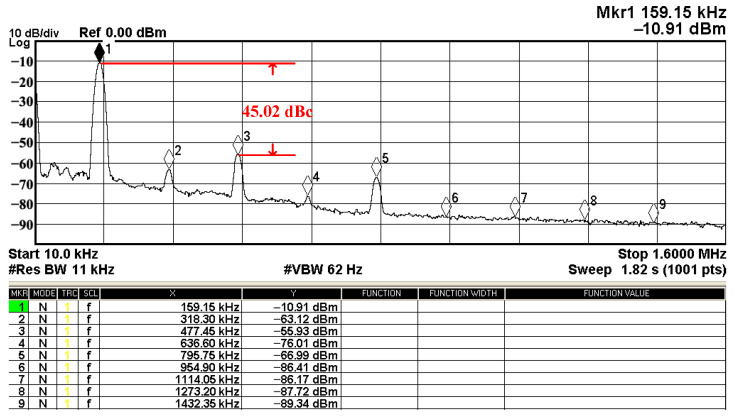
Measured results of the SFDR of the first proposed circuit at V_o1_.

**Figure 48 sensors-22-09379-f048:**
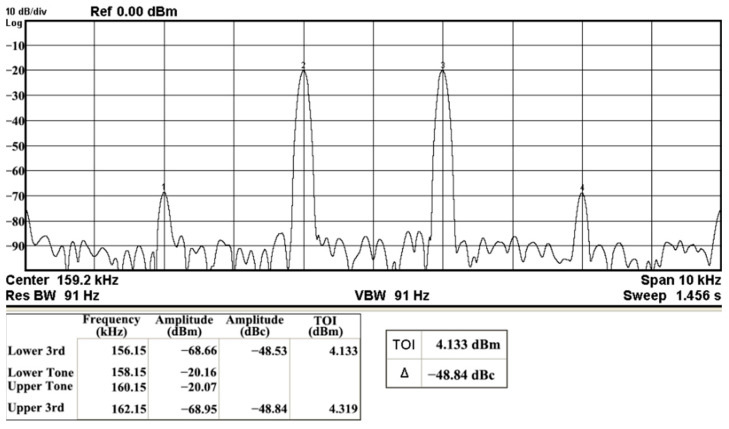
Measured results of the IMD of the first proposed circuit at V_o1_.

**Figure 49 sensors-22-09379-f049:**
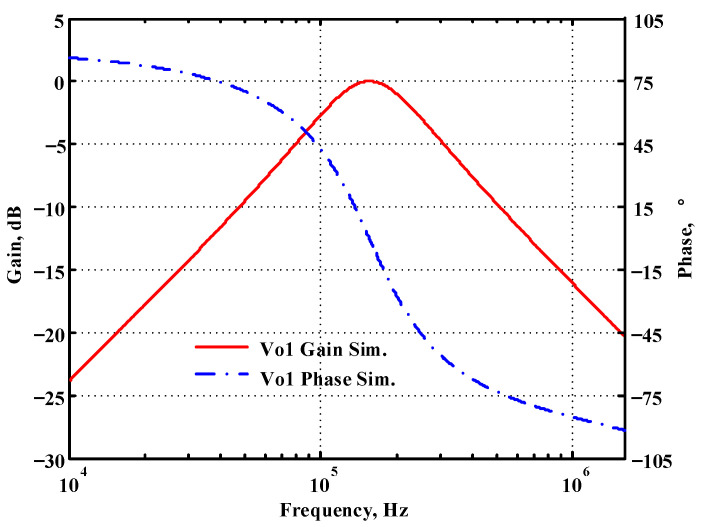
Simulated gain and phase frequency response of the second proposed circuit at V_o1_.

**Figure 50 sensors-22-09379-f050:**
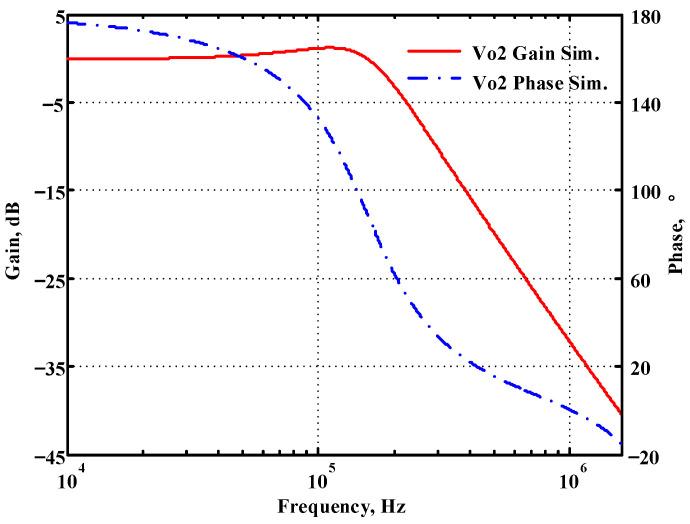
Simulated gain and phase frequency response of the second proposed circuit at V_o2_.

**Figure 51 sensors-22-09379-f051:**
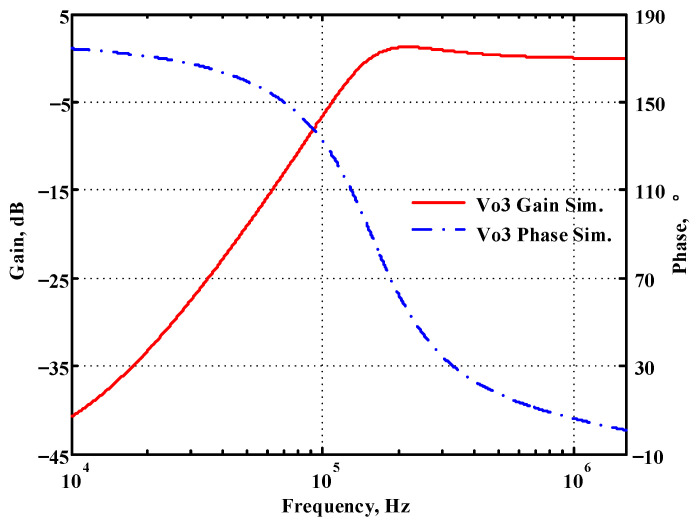
Simulated gain and phase frequency response of the second proposed circuit at V_o3_.

**Figure 52 sensors-22-09379-f052:**
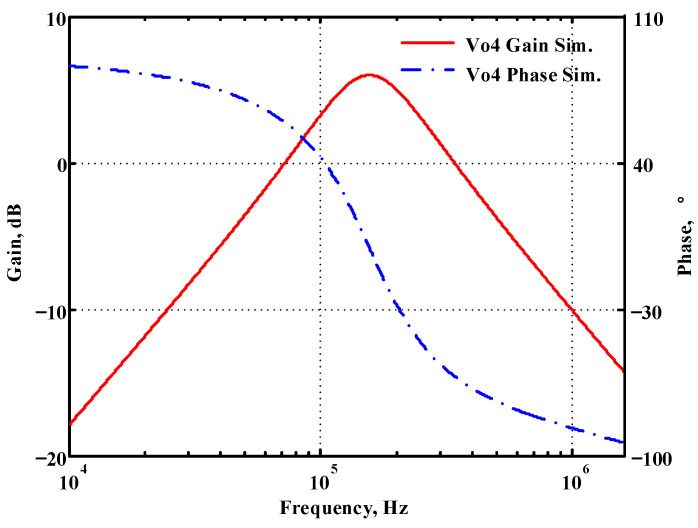
Simulated gain and phase frequency response of the second proposed circuit at V_o4_.

**Figure 53 sensors-22-09379-f053:**
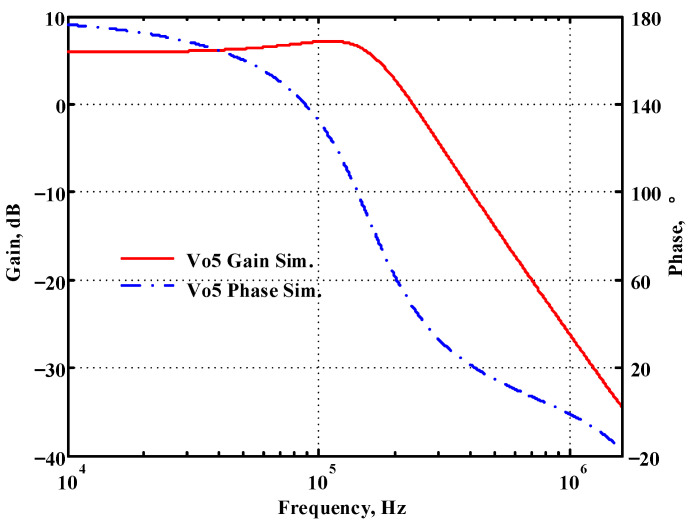
Simulated gain and phase frequency response of the second proposed circuit at V_o5_.

**Figure 54 sensors-22-09379-f054:**
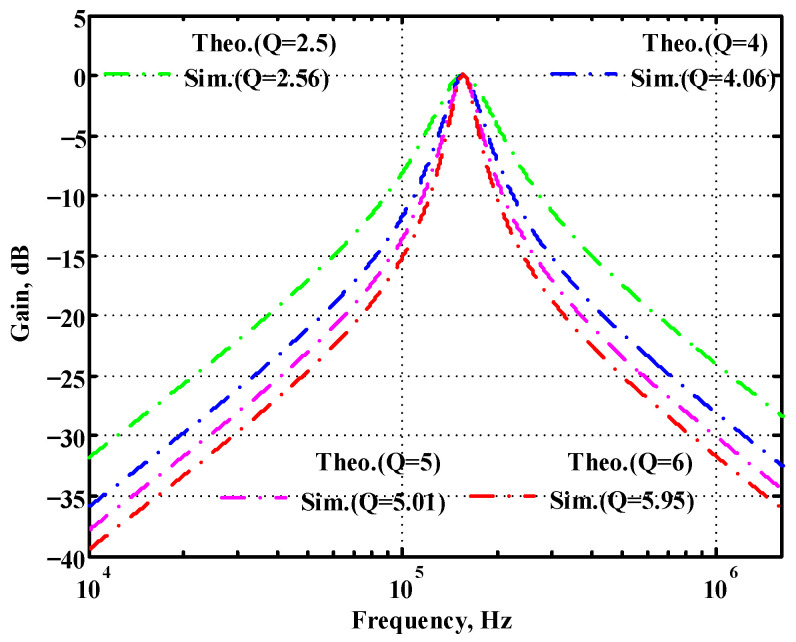
Simulated gain response of the second proposed circuit at V_o1_ by changing Q and keeping f_o_ constant.

**Figure 55 sensors-22-09379-f055:**
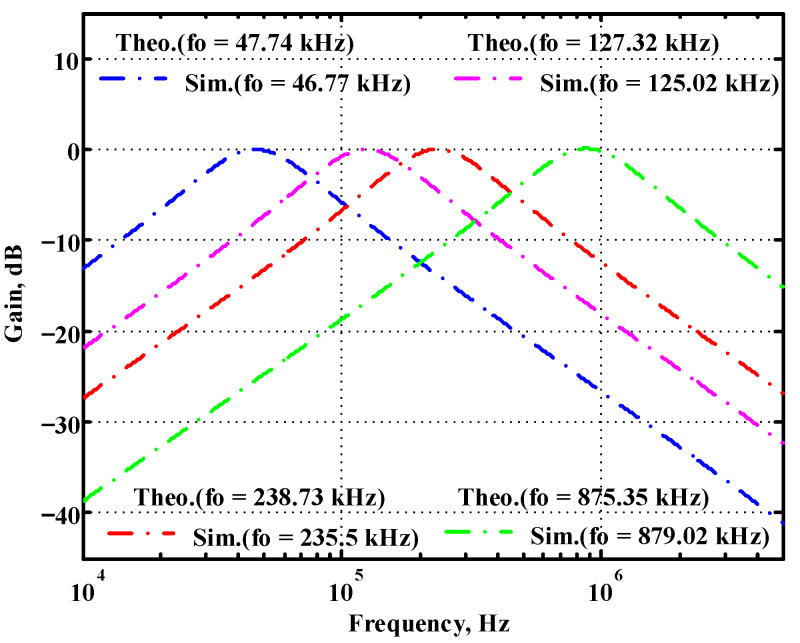
Simulated gain response of the second proposed circuit at V_o1_ by changing f_o_ and keeping Q constant.

**Figure 56 sensors-22-09379-f056:**
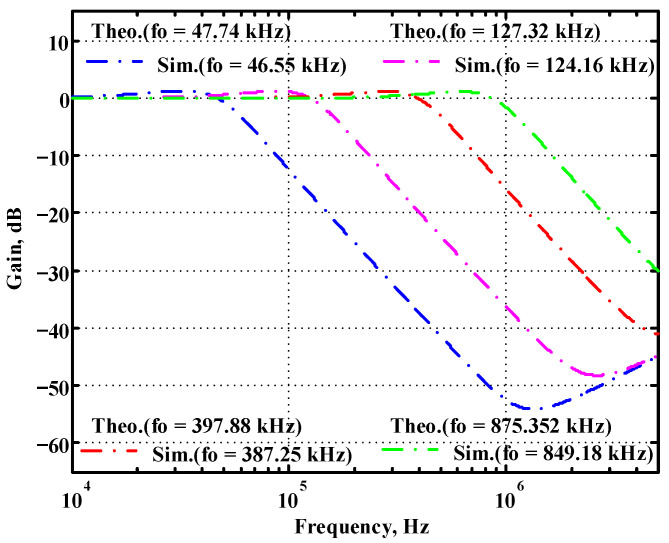
Simulated gain response of the second proposed circuit at V_o2_ by changing f_o_ and keeping Q constant.

**Figure 57 sensors-22-09379-f057:**
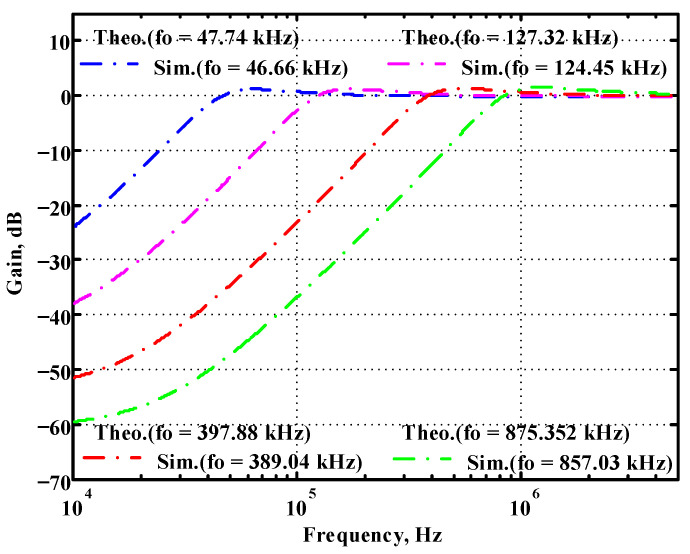
Simulated gain response of the second proposed circuit at V_o3_ by changing f_o_ and keeping Q constant.

**Figure 58 sensors-22-09379-f058:**
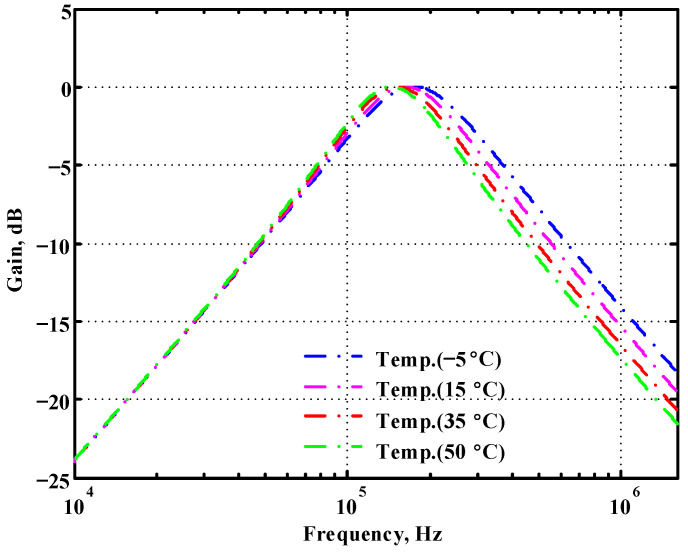
Simulated temperature variation at V_o1_ for the second proposed circuit.

**Figure 59 sensors-22-09379-f059:**
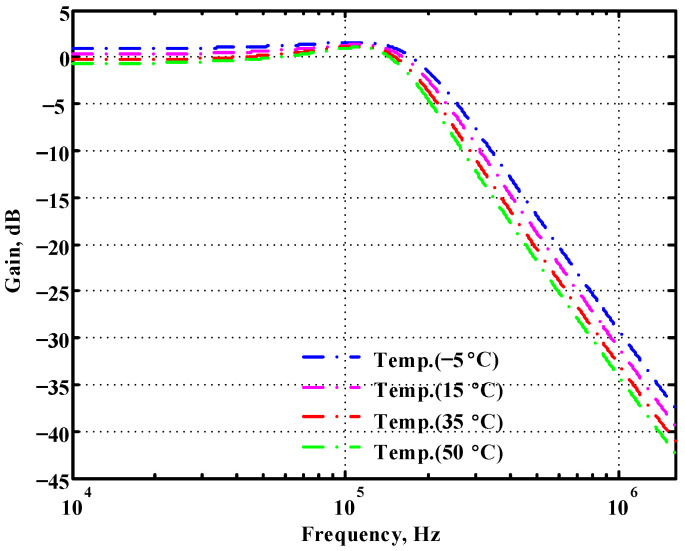
Simulated temperature variation at V_o2_ for the second proposed circuit.

**Figure 60 sensors-22-09379-f060:**
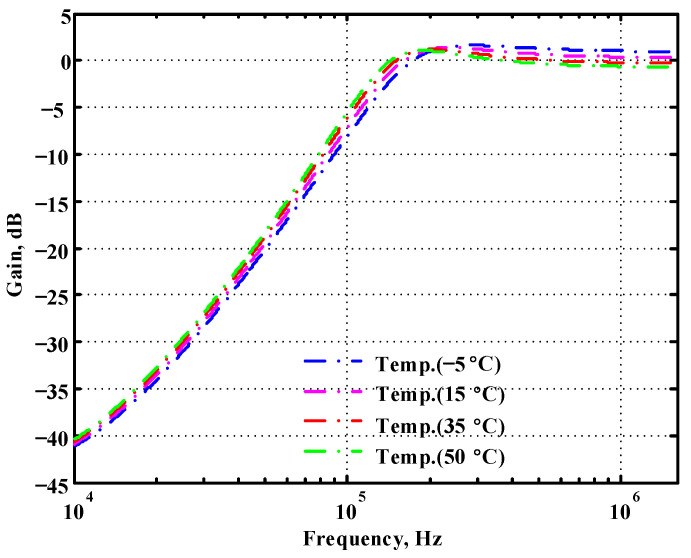
Simulated temperature variation at V_o3_ for the second proposed circuit.

**Figure 61 sensors-22-09379-f061:**
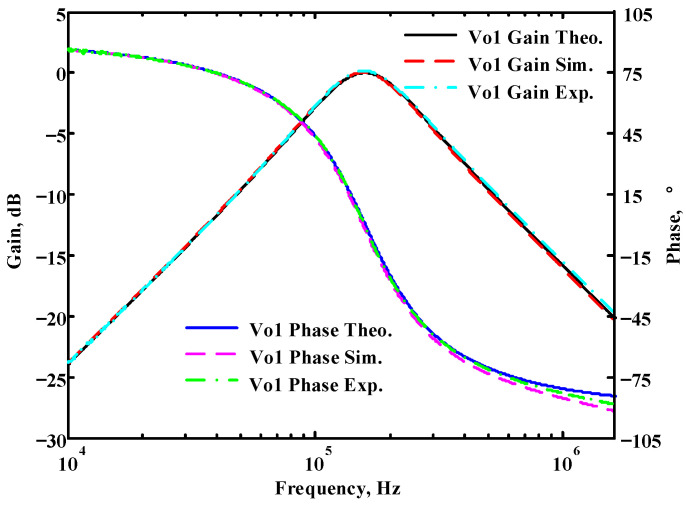
Simulated, measured, and theoretical comparison results of the second proposed circuit at V_o1_.

**Figure 62 sensors-22-09379-f062:**
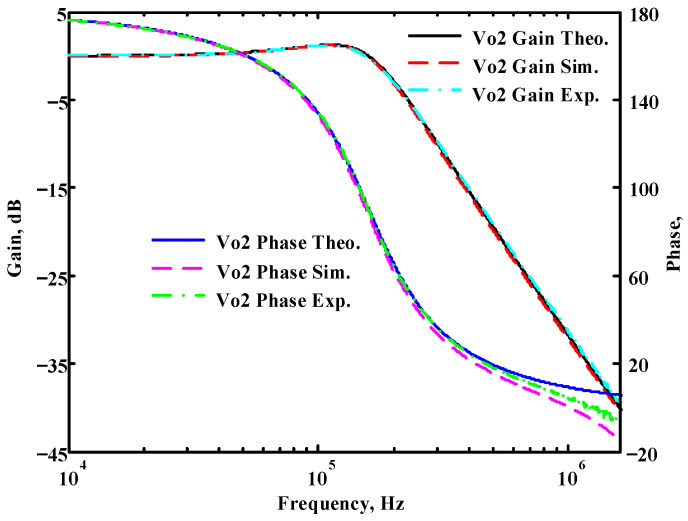
Simulated, measured, and theoretical comparison results of the second proposed circuit at V_o2_.

**Figure 63 sensors-22-09379-f063:**
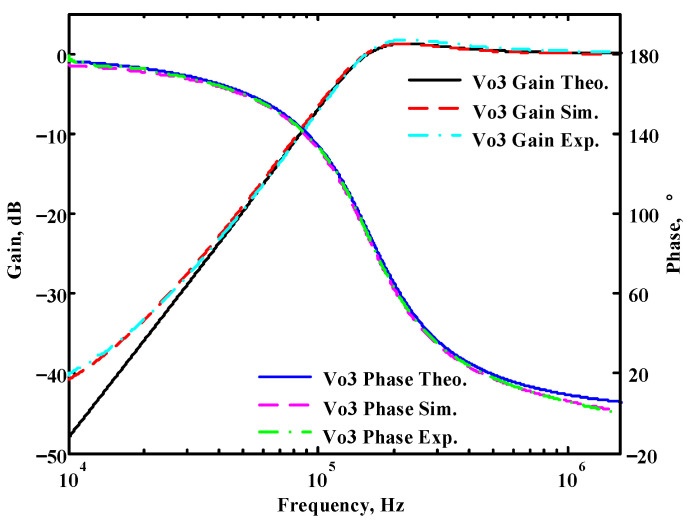
Simulated, measured, and theoretical comparison results of the second proposed circuit at V_o3_.

**Figure 64 sensors-22-09379-f064:**
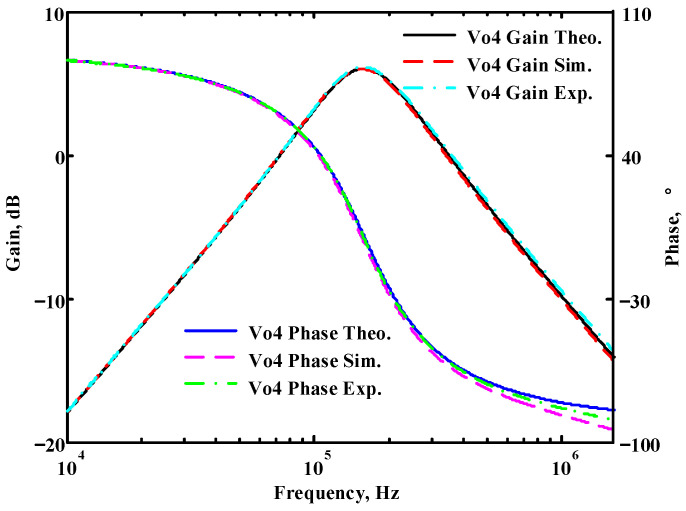
Simulated, measured, and theoretical comparison results of the second proposed circuit at V_o4_.

**Figure 65 sensors-22-09379-f065:**
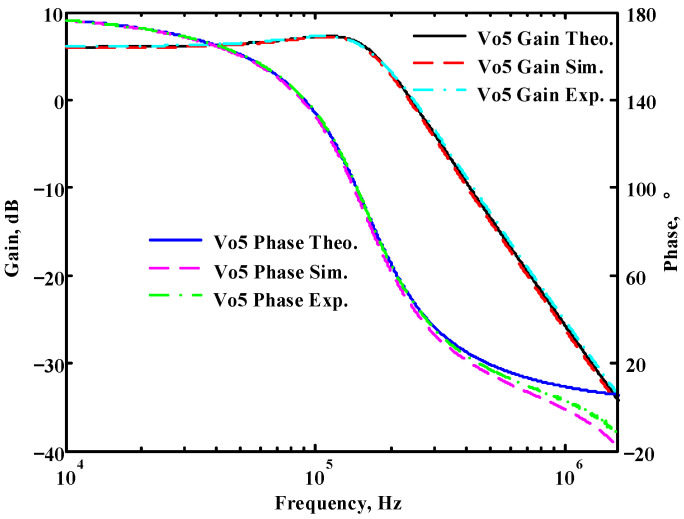
Simulated, measured, and theoretical comparison results of the second proposed circuit at V_o5_.

**Figure 66 sensors-22-09379-f066:**
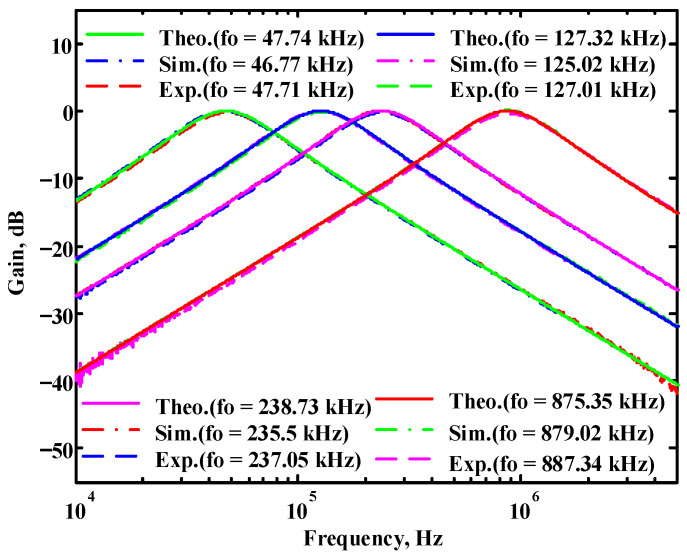
Simulated, measured, and theoretical comparison results of the electronic tunability f_o_ for the second proposed circuit at V_o1_ without affecting the Q value.

**Figure 67 sensors-22-09379-f067:**
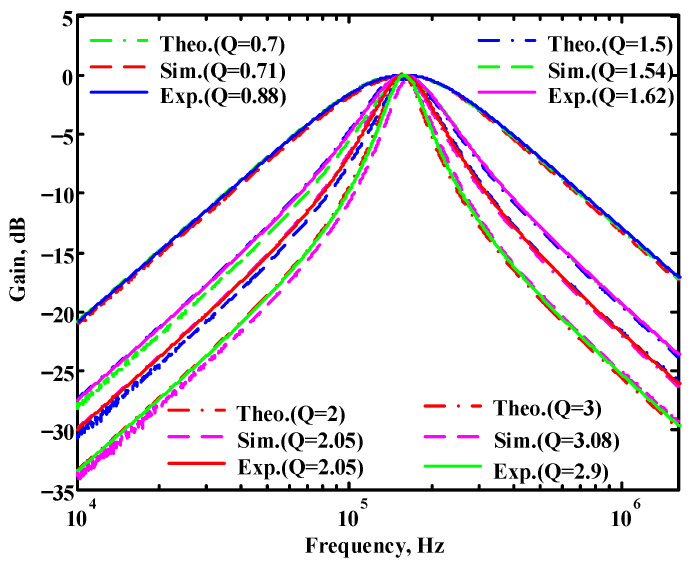
Simulated, measured, and theoretical comparison results of the electronic tunability Q for the second proposed circuit at V_o1_ without affecting the f_o_ value.

**Figure 68 sensors-22-09379-f068:**
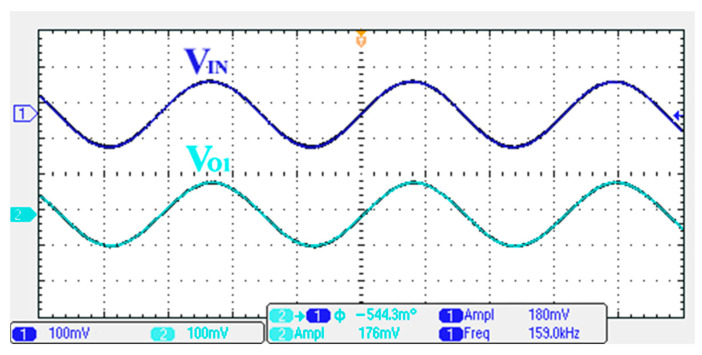
Measured output and input characteristics of the second proposed circuit at V_o1_.

**Figure 69 sensors-22-09379-f069:**
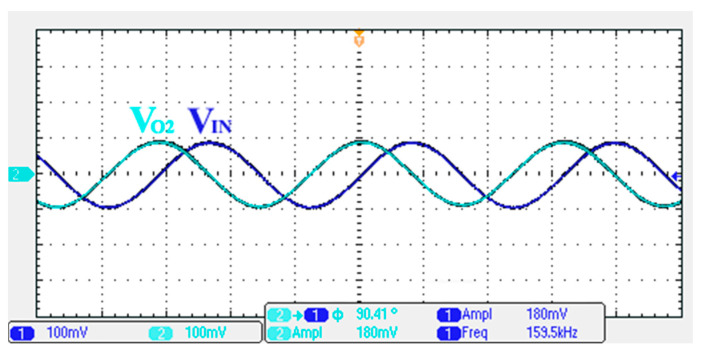
Measured output and input characteristics of the second proposed circuit at V_o2_.

**Figure 70 sensors-22-09379-f070:**
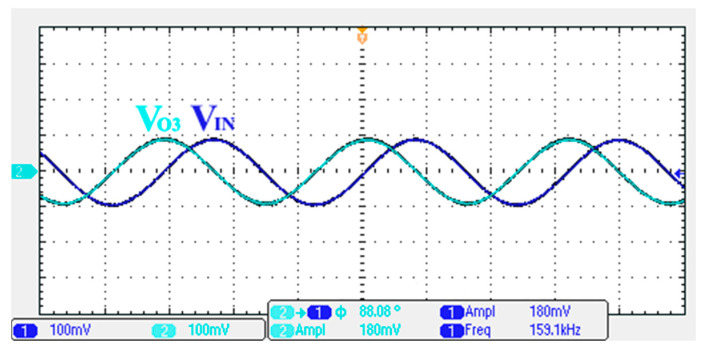
Measured output and input characteristics of the second proposed circuit at V_o3_.

**Figure 71 sensors-22-09379-f071:**
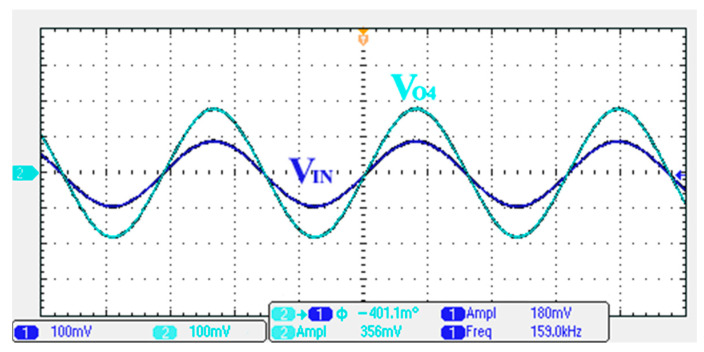
Measured output and input characteristics of the second proposed circuit at V_o4_.

**Figure 72 sensors-22-09379-f072:**
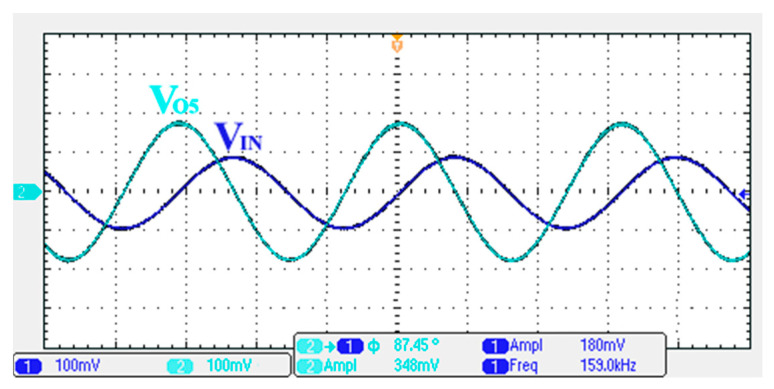
Measured output and input characteristics of the second proposed circuit at V_o5_.

**Figure 73 sensors-22-09379-f073:**
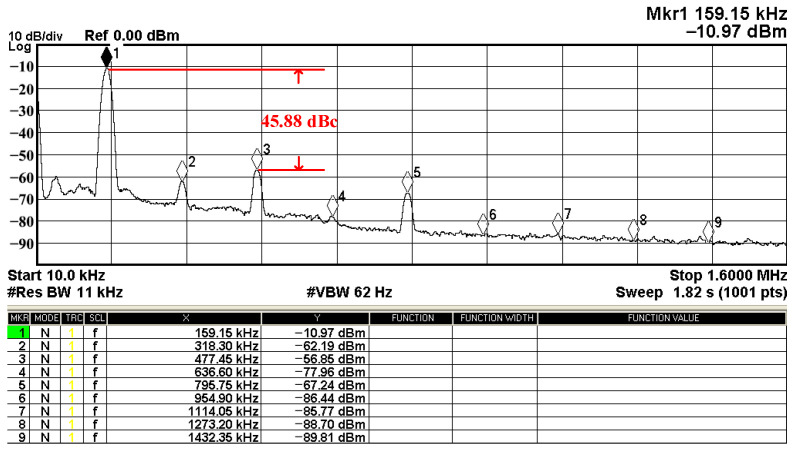
Measured output spectrum of Figure 68.

**Figure 74 sensors-22-09379-f074:**
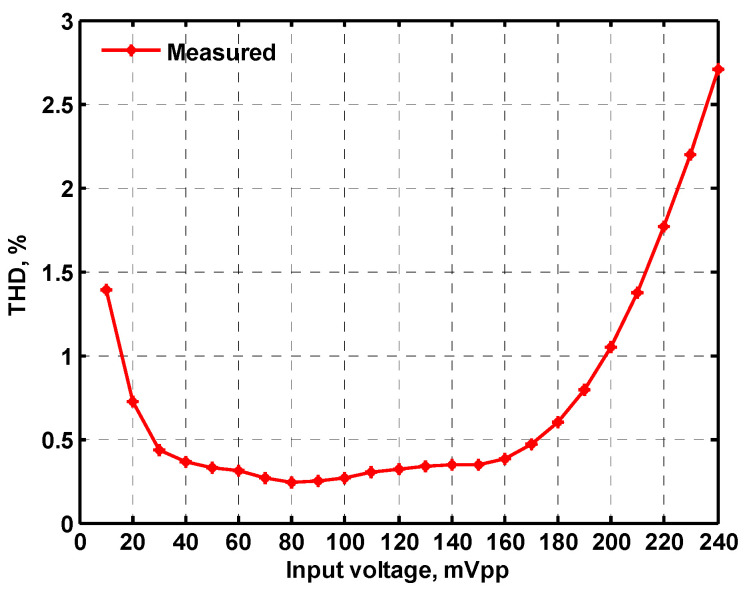
Measured THD of the second proposed circuit at V_o1_ in Figure 6 versus peak-to-peak input voltage signal at 159.15 kHz.

**Figure 75 sensors-22-09379-f075:**
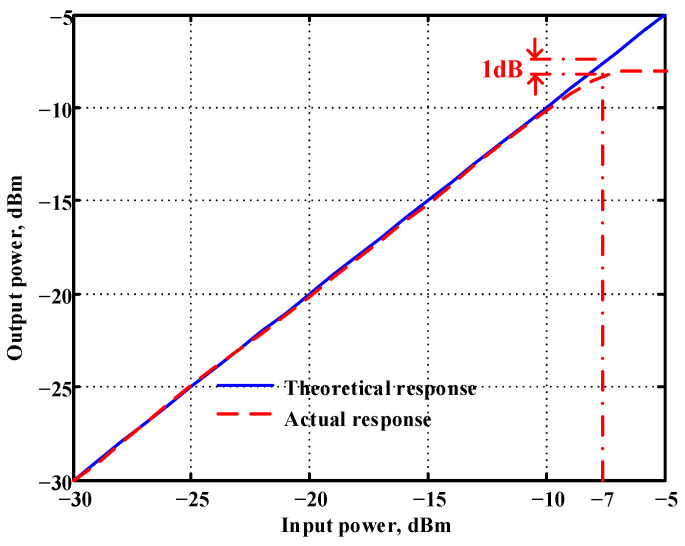
Measured results of the P1dB point of the second proposed circuit at V_o1_.

**Figure 76 sensors-22-09379-f076:**
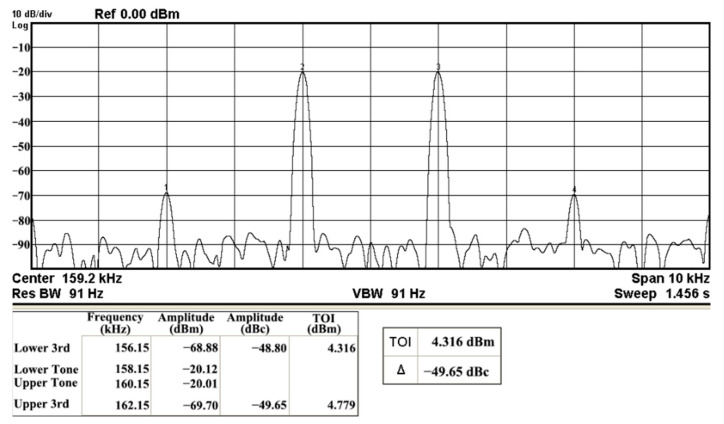
Measured results of the IMD3 of the second proposed circuit at V_o1_.

**Table 1 sensors-22-09379-t001:** Comparison of recent electronically tunable VM second-order multifunction filter specifications.

Ref.	ABB and Passive Elements Used		No. of Commercial ICs Realized	No. of Filtering Functions at the Same Time	No. of Low Output Impedance Filtering Functions Used	Only High-Input Impedance Used	Only Two Grounded Capacitors Used	Simultaneous Realization of Filtering Functions	Independent Gain-Controlled Filtering Functions	Orthogonal Electronic Control of Q and f_o_	Independent Electronic Control Q without Affecting f_o_	Q-Value or Q Tuning Range	Pole Frequency f_o_ or f_o_ Tuning Range (kHz)
	ABB	Passive											
[22]	4 CCII	2C + 4R	4	5	0	yes	yes	LPF, BPF, HPF, BRF, APF ^a^	no	no	no	<5	15
[23]	3 CFA	2C + 4R	3	3	3	yes	yes	LPF, BPF, BRF	no	no	no	1.1–4	49–197
[24]	3 CFA	2C + 4R	3	3	3	yes	yes	LPF, BPF, HPF	no	no	no	0.6–1.6	86–170
[25]	4 CFA in filter 1a	2C + 5R + 2SW	4	4	4	yes	yes	LPF, BPF, BRF, HPF/APF	no	no	no	0.3–3.2	150
[25]	4 CFA in filter 1b	2C + 6R + 2SW	4	4	4	yes	yes	LPF, BPF, BRF, HPF/APF	no	no	no	<0.7	22–210
[25]	4 CFA in filter 1c	2C + 6R + 1SW	4	4	4	yes	yes	LPF, BPF, BRF, HPF/APF	no	no	no	0.3–3.3	22–210
[26]	3 VDDDA	2C + 1R	5	6	3	yes	yes	LPF, 2 BPF, HPF, BRF, APF	no	yes	yes	0.3–26.6	1047
[27]	3 VDDDA	2C + 1R	5	5	2	yes	yes	LPF, BPF, HPF, BRF, APF	no	yes	yes	1–4.5	625–1500
[28]	2 VD-DIBA	2C + 2R	4	4	2	yes	yes	LPF, BPF, HPF, BRF	no	no	no	2–5	66–144
[29]	5 OTA	2C	5	3	0	yes	yes	LPF, BPF, BRF	no	yes	yes	1–3	156–472
[30]	5 OTA	2C	5	3	0	yes	yes	LPF, BPF, BRF	no	yes	yes	1–2.9	107–284
[31]	4 OTA	2C	5	3	0	yes	yes	LPF, BPF, BRF	no	yes	yes	1–1.7	106–213
[32]	3 LT1228 in filter 2a	2C + 3R	3	3	2	yes	yes	3 BPF	1 BPF	no	no	1–2.9	161
[32]	3 LT1228 in filter 3a	2C + 3R	3	3	2	yes	yes	3 BPF	1 BPF	no	no	1–2.9	165
[33]	3 LT1228	2C + 4R	3	3	2	no	yes	HPF, BPF, LPF	no	yes	yes	0.4–1.7	100–373
[34]	2 LT1228 in filter 1	2C + 3R	2	3	2	no	yes	HPF ^a^, BPF ^a^, LPF ^a^	no	no	no	<1.6	135–826
[34]	2 LT1228 in filter 2	2C + 3R	2	3	2	no	yes	HPF ^a^, BPF ^a^, LPF ^a^	no	no	no	<1.6	44.3–275
[35]	2 LT1228 in filter 2a	2C + 5R	2	3	2	no	yes	HPF ^a^, BPF ^a^, LPF ^a^	no	no	no	0.9–4.5	57–205
[35]	2 LT1228 in filter 2b	2C + 5R	2	3	2	no	yes	HPF ^a^, BPF ^a^, LPF ^a^	no	no	no	0.9–4.5	99
Circuit 1	3 LT1228	2C + 5R	3	5	3	yes	yes	HPF, 2 BPF, 2 LPF	1 LPF, 1 BPF	yes	yes	0.7–6	46–961
Circuit 2	3 LT1228	2C + 5R	3	5	3	yes	yes	1 HPF, 2 BPF, 2 LPF	1 LPF, 1 BPF	yes	yes	0.7–6	46–879

Note: ABB: active building block; ^a^ requirements for resistor matching conditions.

**Table 2 sensors-22-09379-t002:** Time domain characteristics of the output phase errors of the first proposed filter measured at 159.15 kHz.

Output Terminal	Operating Pole Phase		Output Phase Error
	Theoretical	Measured	
V_o1_	0°	−0.143°	−0.143°
V_o2_	−90°	−90.97°	−0.97°
V_o3_	−90°	−90.27°	−0.27°
V_o4_	0°	−0.601°	−0.601°
V_o5_	−90°	−90.13°	−0.13°

**Table 3 sensors-22-09379-t003:** Frequency-domain characteristics of the pole frequency error measured at ideal pole phase for the first proposed circuit.

Output Terminal	Filter Pole Frequency			Percentage Error of the Pole Frequency	
	Theoretical	Simulated	Measured	Simulated	Measured
V_o1_	159.15 kHz	155.59 kHz	157.2 kHz	−2.23%	−1.22%
V_o2_	159.15 kHz	155.23 kHz	158.37 kHz	−2.46%	−0.49%
V_o3_	159.15 kHz	155.59 kHz	157.58 kHz	−2.23%	−0.98%
V_o4_	159.15 kHz	155.23 kHz	158.38 kHz	−2.46%	−0.48%
V_o5_	159.15 kHz	154.88 kHz	158.76 kHz	−2.68%	−0.24%

**Table 4 sensors-22-09379-t004:** Summary of the measured performance of the first proposed VM LT1228-based second-order multifunction filter.

Factor							
Power Supply (V)	PD (W)	Pole Frequency (kHz)	Pole Frequency Tuning Range (kHz)	P1dB (dBm)	IMD3 (dBc)	TOI (dBm)	SFDR (dBc)
±15	0.69	157.2	47.2–951.94	−7.1	−48.84	4.133	45.02

**Table 5 sensors-22-09379-t005:** Frequency-domain characteristics of the pole frequency error measured at ideal pole phase for the second proposed circuit.

Output Terminal	Filter Pole Frequency			Percentage Error of the Pole Frequency	
	Theoretical	Simulated	Measured	Simulated	Measured
V_o1_	159.15 kHz	155.59 kHz	157.55 kHz	−2.23%	−1%
V_o2_	159.15 kHz	155.23 kHz	158.22 kHz	−2.46%	−0.58%
V_o3_	159.15 kHz	155.59 kHz	156.67 kHz	−2.23%	−1.55%
V_o4_	159.15 kHz	155.23 kHz	157.71 kHz	−2.46%	−0.9%
V_o5_	159.15 kHz	154.88 kHz	158.98 kHz	−2.68%	−0.1%

**Table 6 sensors-22-09379-t006:** Time-domain characteristics of the output phase error measured for the second proposed circuit at an operating pole frequency of 159.15 kHz.

Output Terminal	Operating Pole Phase		Output Phase Error
	Theoretical	Measured	
V_o1_	0°	−0.544°	−0.544°
V_o2_	90°	90.41°	0.41°
V_o3_	90°	88.08°	−1.92°
V_o4_	0°	−0.401°	−0.401°
V_o5_	90°	87.45°	−2.55°

**Table 7 sensors-22-09379-t007:** Summary of the measured performance of the second proposed VM LT1228-based second-order multifunction filter.

Factor							
Power Supply (V)	PD (W)	Pole Frequency (kHz)	Pole Frequency Tuning Range (kHz)	P1dB (dBm)	IMD3 (dBc)	TOI (dBm)	SFDR (dBc)
±15	0.69	157.55	47.71–887.34	−7	−49.65	4.316	45.88

## Data Availability

Not applicable.

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
