# Peer review of "Synthesis of High-Input Impedance Electronically Tunable Voltage-Mode Second-Order Low-Pass, Band-Pass, and High-Pass Filters Based on LT1228 Integrated Circuits"

_sensors, 2022, doi:10.3390/s22239379_

Round 1

Reviewer 1 Report

This paper proposes two new syntheses of high-input impedance electronically tunable voltage-mode low-pass, band-pass, and high-pass filters based on LT1228s. This is a very good and practical idea. Compared with the recently proposed technique in [32], these two proposed filters have more advantages, such as high-input and low-output impedances, orthogonal/independent electronic tuning capability, independent gain control low-pass and band-pass filtering responses, and synthesis filter topology analysis methods. The manuscript is well prepared, clearly explained and experimentally verified. The following two comments can be considered when revising the study version.
1. The second proposed circuit provides time domain measurement of Vo1 only, while the other Vo2, Vo3, Vo4 and Vo5 can be considered for measurements.
2. The phase error between the output and input waveforms measured in the time domain can be summarized in a table.

Author Response

The authors would like to express his gratitude to anonymous reviewers for carefully reviewing the paper, for many thoughtful comments in the original manuscript. The manuscript has been revised and improved according to the suggestions of reviewers. The changes in the revised manuscript are highlighted in yellow.

Comments and Suggestions for Authors

This paper proposes two new syntheses of high-input impedance electronically tunable voltage-mode low-pass, band-pass, and high-pass filters based on LT1228s. This is a very good and practical idea. Compared with the recently proposed technique in [32], these two proposed filters have more advantages, such as high-input and low-output impedances, orthogonal/independent electronic tuning capability, independent gain control low-pass and band-pass filtering responses, and synthesis filter topology analysis methods. The manuscript is well prepared, clearly explained and experimentally verified.

Author response: Thank you very much for your appreciation.

Concern # 1: The second proposed circuit provides time domain measurement of Vo1 only, while the other Vo2, Vo3, Vo4 and Vo5 can be considered for measurements.

Author response:

Thanks for your comment. We have added the time domain measurements of the second proposed circuit in Figures 69 to 72 of the revised manuscript. (Please see Section 3 Simulation and Experimental results on page 33, from line 570 and on page 40, from line 644 of the revised manuscript.)

Concern # 2: The phase error between the output and input waveforms measured in the time domain can be summarized in a table.

Author response:

Thanks for your comment. We have summarized the phase errors between the output and input waveforms in Table 6 of the revised manuscript. (Please see Section 3 Simulation and Experimental results on page 33, from line 572 and on page 41, from line 651 of the revised manuscript.)

Reviewer 2 Report

This paper introduces two new high-input impedance electronically tunable voltage mode (VM) multifunction second-order architectures with band-pass (BP), low-pass (LP) and high pass (HP) filters. I think the structure is interesting. However, some format and grammar errors. 

Author Response

The authors would like to express his gratitude to anonymous reviewers for carefully reviewing the paper, for many thoughtful comments in the original manuscript. The manuscript has been revised and improved according to the suggestions of reviewers. The changes in the revised manuscript are highlighted in yellow.

Comments and Suggestions for Authors

This paper introduces two new high-input impedance electronically tunable voltage mode (VM) multifunction second-order architectures with band-pass (BP), low-pass (LP) and high pass (HP) filters. I think the structure is interesting.

Author response: Thank you very much for your appreciation.

Concern # 1: However, some format and grammar errors.

Author response:

Thanks for your comment. We have reviewed and corrected formatting and grammar errors in the revised manuscript.

Reviewer 3 Report

This paper explores the design of electronically tunable active filters.  It is interesting to see that several factors, such as the Q factor and the f0 etc, can be adjusted independently, which provides freedom to the adoption in the sensor system. Therefore, from this perspective, the work could be interesting to the readers of the journal. However, the current version is more like a report rather than an academic paper and thus needs to be revised. In addition, several comments are given below for the author to improve the readability further.

Major:

1.     The authors should provide the tuning range of the parameters, such as the Q factor, and provide some comparison with existing designs in Table I. In addition, the explanation with simulation what the limiting factor for the boundaries is preferable.

2.  Simulation details should be given in the manuscript.

3.  The authors should show the stability at different temperatures.

4. Can the authors demonstrate how effective the filter is when used as the filter by interfacing with certain sensors?

Minor:

5.     What is the difference between Fig.9 & Fig.10?

6. Figures should be plotted professionally.

Author Response

The authors would like to express his gratitude to anonymous reviewers for carefully reviewing the paper, for many thoughtful comments in the original manuscript. The manuscript has been revised and improved according to the suggestions of reviewers. The changes in the revised manuscript are marked in blue. We provide a response letter detailing point-by-point revisions to the manuscript and responses to reviewer comments. Please see the attached response.

Reviewer 4 Report

The authors have designed and implemented two all-pass filters using some commercial integrated circuits. The initial idea, and the method reaching the results are acceptable. The authors have started from equations and finally have gained the filter structures. However, there are some concerns they should clarify, as follows:

1-     Are the gains of the filters variable and controllable? Please clarify and put comment on that.

2-     Please describe the benefits of the proposed circuits based on the comparison Table 1, in detail.

3-     Is there any possibility to fabricate your designs in CMOS technology with more advantages?

4-     Please report the THD of the second proposed circuit like Fig. 16.

5-     Is the frequency of the LP and HP outputs adjustable? Why there is no results for them? Why do not you put figures 37-41 in one Fig?

6-     The simulation results are also required and should be compared with measured results in a Table.

7-     In the introduction, a short discussion should be allocated to MOS-only filters such as the following reference: An Adjustable Dual-Output Current Mode MOSFET-Only Filter," in IEEE Transactions on Circuits and Systems II: Express Briefs, vol. 68, no. 6, pp. 1817-1821, June 2021.

Author Response

(The authors gave the same response as above.)

Round 2

Reviewer 3 Report

thanks for the authors' reply to my questions. i think the manuscript is clear and ready to be accepted.

Reviewer 4 Report

I have no more comment.